# Coupling between spatial compartments integrates morphogenetic patterning in the organ of Corti

Anubhav Prakash[1,2*], Sukanya Raman[1], Raman Kaushik[1], Pallavi Manchanda[1], Anton S. Iyer[1], Raj K. Ladher[1*]

1 National Centre for Biological Sciences, Tata Institute of Fundamental Research, Bangalore, India,
2 Trivedi School of Biosciences, Ashoka University, Plot No. 2, Rajiv Gandhi Education City, National Capital Region P.O. Rai, Sonipat Haryana, India

* anubhav.prakash@ashoka.edu.in (AP); rajladher@ncbs.res.in (RKL)

## Abstract

Morphogenetic information arises from a combination of genetically encoded cellular properties and emergent cellular behaviors. The spatio-temporal implementation of this information is critical to ensure robust, reproducible tissue shapes, yet the principles underlying its organization remain unknown. We investigated this principle using the mouse auditory epithelium, the organ of Corti (OC). OC consists of a sensory domain, which transduces sound through polar mechanosensory hair cells (HC), part of a mosaic with supporting cells (SC). On either side of the sensory domain are non-sensory domains. These domains undergo cellular rearrangements, which, together, lead to a spiral cochlea that contains planar polarized HCs. This makes the mammalian cochlea a compelling system to understand coordination across spatial scales. Using genetic and ex vivo approaches, we found patterning of OC into sensory and non-sensory domains is associated with a combinatorial expression of adhesion molecules, which underpins OC into spatially defined compartments, enabling planar cell polarity (PCP) cues to regulate compartment-specific organization. Through compartment-specific knockouts of the PCP protein, Vangl2, we find evidence of compartment coupling, a non-linear influence on the organization within one compartment when cellular organization is disrupted in another. In the OC, compartment coupling originates from vinculin-dependent junctional mechanics, coordinating cellular dynamics across spatial scales.

## Introduction

The reproducibility and robustness of morphogenesis result from carefully implemented developmental instructions. Through a combination of signaling cues, gene networks, and self-organizing mechanisms, patterning divides an organogenic field into compartments [1,2]. While compartments allow the segregation of distinct cell

**Data availability statement:** Codes related to this work is available at https://github.com/antoniNCBS/Compartment-coupling-integrates-patterning-and-morphogenetic-information-during-development. Underlying Data is available in S1 Data.

**Funding:** This work was supported by the Department of Atomic Energy, Government of India, Project Identification No. RTI 4006, and grants from SERB (CRG/2018/001235), Infosys Foundation, TIFR Infosys-Leading Edge Grant, the Royal National Institute for Deaf People (G97) to RKL. AP was supported by the International Foundation for Research and Education, via a Simons-Ashoka ECF fellowship. The sponsors or funders played no role in the study design, data collection and analysis, decision to publish, or preparation of the manuscript.

**Competing interests:** The authors have declared that no competing interests exist.

**Abbreviations :** ACRC, Animal Care and Resource Centre; BLBP, brain lipid binding-protein; BSA, Bovine Serum Albumin; CE, convergence and extension;E, embryonic day; HC, hair cells; HnC, Hensen's cells; IHC, inner HCs; IPhC, Inner Phalangeal cells; KO, Kölliker's organ; IKO, lateral KO; mKO, medial edge of the KO; NA, numerical aperture; NM, non-muscle myosin; OC, organ of Corti; OHC, outer HCs; PBS, Phosphate-Buffered Saline; PCP, planar cell polarity; PCR, polymerase chain reaction; PFA, paraformaldehyde; RLC, regulatory light chain; SC, supporting cells.

behaviors, these behaviors must be spatiotemporally coordinated across compartments so that functional organs, with the correct shape and pattern, forms. The mouse auditory epithelia is an excellent system to investigate this unknown coordination.

The mouse auditory epithelium, called the organ of Corti (OC), is found within the cochlea. It is a spiral-shaped organ and is responsible for detecting and transducing sound across a wide spectrum of frequencies [3,4]. Sound is transduced by hair cells (HC) through an asymmetric hearing organelle, the hair bundle on their apical surface. HCs are of two types: inner HCs (IHC) arranged into a single row that transmit information to the brain; and outer HCs (OHC) arranged into three rows that amplify the mechanical input. Both these HCs are intercalated by supporting cells (SC) and together form the sensory domain of OC. This sensory domain is flanked by non-sensory compartments. On the medial side (inner edge of spiral) is Kölliker's organ (KO) and on the lateral side (outer edge of spiral) is a lateral non-sensory compartment, which includes Hensen's and Claudius' cells (Figs 1A and S1A).

The cochlea is initially apparent as a ventral out-pocketing of the otocyst. Radial patterning of the nascent cochlear duct by morphogen signaling establishes non-sensory and sensory domains of the OC [5,6]. The cells in the sensory domain become post-mitotic and, through juxtacrine signaling, differentiate into a mosaic of HC and SC [7,8]. As HC develop, they form asymmetrically localized hair bundles which align with the tissue axis, a process known as planar polarity. Both experimental approaches and mathematical modeling have shown that local coordination among HCs and SCs driven by differential junctional tension could drive the organization of HCs and align HC polarity to the tissue axis [9–12]. While cells locally coordinate in the sensory domain, large-scale convergence and extension (CE) movements together with growth and proliferation cause the OC to elongate contributing to morphogenesis and the spiraling of the OC [13–18]. Thus, local domain-specific processes that order HCs must integrate with large-scale tissue remodeling. How they integrate is not known.

The OC compartmentalizes into smaller domains that show a combinatorial expression of Cadherin 1, 2, and 4 [15,19,20]. Using mutants of fibroblast growth factor signaling and ex vivo cultures, we find that the adhesion code ensures compartment integrity during convergent and extension movements. Each compartment uses the planar cell polarity (PCP) molecule, Vangl2, to develop a distinct cellular organization. Using compartment-specific knockouts of Vangl2, we find that cellular organization within each compartment has a non-linear influence on the organization of another compartment, a novel phenomenon called compartment coupling. In mice mutant for the junctional force transmission component, Vinculin, we show that compartment coupling has a mechanical element. Our work suggests that compartment coupling underpins the integration of local cellular ordering with large-scale tissue remodeling. Given the widespread use of compartments, inter-compartment coupling is likely a fundamental feature of the morphogenesis of many developing tissues and organs.

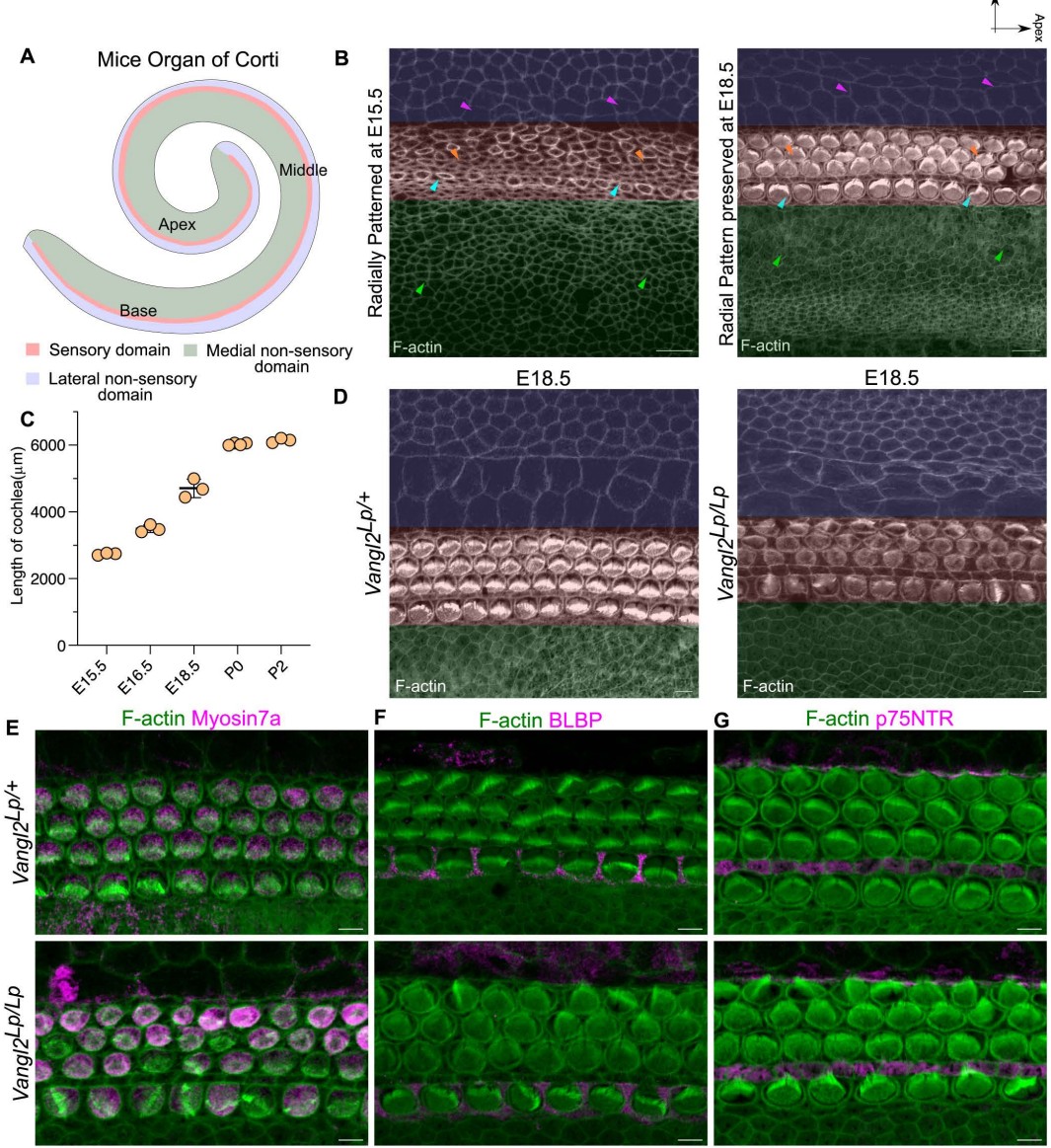

**Fig 1. Cochlea undergoes pattern-preserving convergent extension movement during development. (A)** Schematic of spiral-shaped mouse organ of Corti (OC) representing Kölliker's organ (KO) (medial non-sensory domain), sensory domain, and lateral non-sensory domain. **(B)** Base part of OC from E15.5 and E18.5 stained for F-actin. Green overlay indicates KO domain, red overlay indicates sensory domain, and blue overlay indicates lateral non-sensory domain. Blue and orange arrowhead indicates medial and lateral sensory domain, respectively. Magenta arrowheads indicate Hensen's cell and green arrowheads shows cells in KO domain. **(C)** Length of cochlea from E15.5 to post-natal day 2. $N=3$ embryos for each stage. **(D)** Base of E18.5 OC from heterozygous (*Vangl2 Lp/+*) and homozygous (*Vangl2 Lp/Lp*) *looptail* mutant stained for F-actin (gray). Overlay indicates domains as **B.** $N=8$. **(E)** Base of E18.5 OC from heterozygous (*Vangl2 Lp/+*) and homozygous (*Vangl2 Lp/Lp*) *looptail* mutant stained for F-actin (green) and Myosin 7a (magenta). $N=4$. **(F)** Base of E18.5 OC from heterozygous (*Vangl2 Lp/+*) and homozygous (*Vangl2 Lp/Lp*) *looptail* mutant stained for F-actin (green) and BLBP (magenta). $N=4$. **(G)** Base of E18.5 OC from heterozygous (*Vangl2 Lp/+*) and homozygous (*Vangl2 Lp/Lp*) *looptail* mutant stained for F-actin (green) and p75NTR (magenta). $N=4$. Scale Bar: 5 µm in E–G, and 10 µm in B and **D.** Image orientation: Top is lateral, Right is Apex. Underlying data available in S1 Data.

## Results

**Domain organization is preserved during cochlear elongation.** To investigate the mechanism that could integrate local organization with tissue-scale remodeling, we first asked how the organization in OC evolves from embryonic day (E)15.5, when the HCs are first apparent, to E18.5, when the HCs achieve their final organization. Immunostaining for a marker of HCs, Myosin 7a, shows the sensory domain is already established by E15.5 (S1B Fig). Positive immunostaining for p75NTR, a molecular marker for inner pillar cells, which segregates the IHCs from OHCs, shows the presence of medial and lateral sensory domains at this stage (S1C Fig). Further, the expression of brain lipid binding-protein (BLBP), a molecular marker for Hensen's cells (HnC, a SC type lateral to OHC) and Inner Phalangeal cells (IPhC, a SC type intercalating IHC) shows that the sensory and the non-sensory domains are established by E15.5 (S1D Fig). This organization of OC into medial and lateral non-sensory and sensory domains suggests that by E15.5, the OC is radially patterned (Fig 1B).

From E15.5 to E18.5, the cochlea elongates from 2,734±33 µm to 4,702±27 µm (Figs 1C and S1E). Previous studies have shown that cell growth, migration, intercalation, and tissue-scale convergent-extension movements drive this elongation [13–18]. Such movements are expected to disrupt the organization established at E15.5 [21]. However, they do not. Immunostaining of the E18.5 OC for Myosin 7a, BLBP, and p75NTR revealed that domain organization was maintained during CE-mediated cochlear elongation (Figs 1B and S1B–S1D).

While previous studies have investigated the organization of the medial and lateral sensory domains when CE movements are perturbed, non-sensory domain organization is unclear. Mice mutant for the core PCP protein, Vangl2, show defects in HC PCP and convergence and extension (CE) movements [22–25]. We thus assessed the organization of non-sensory domains in these mutants. Homozygous *looptail* mutants of Vangl2 (referred to as *Vangl2*$^{Lp/Lp}$) have cochlea 2/3rd the length of littermate controls (*Vangl2*$^{Lp/+}$ 5,010±41 µm and *Vangl2*$^{Lp/Lp}$ 3,238±185 µm) (S2A and S2B Fig). Immunostaining for molecular markers for HCs, IPhCs, inner pillar cells, and the distinction in the morphological features of non-sensory cell types (Fig 1D–1G) revealed that the relative position of cell types and domain organization is maintained in the *Vangl2*$^{Lp/Lp}$ mutants with defects in convergent extension (S2C–S2E Fig). This suggests a mechanism to maintain the integrity of individual domains during cochlear elongation.

**Adhesion code defines compartments in the OC.** Studies of mixed cultures of adhesion-molecule expressing cells, on gastrulating amphibian embryos, and the fish neural tube have shown that the differential expression of adhesion molecules could segregate cells into domains [26–29]. The differential expression of cadherins and nectins has been described in sensory domain of the OC, [9,15,20,30,31] leading us to characterize their expression across the entire OC. We focused on the expression of 3 cadherin family members, Cdh1, Cdh2 and Cdh4 (E-, N- and R- cadherin, respectively), as well as Nectin-1 and -2, and the tight junction protein ZO-1.

At E18.5, ZO-1 and Nectin-2 are localized on the apical junctions of all the cells of the OC (S3A and S3B Fig). Nectin-1 is expressed only on the HC-SC junctions (S3C Fig). The expression of Cdh1 was found on the junctions of cells in the lateral domain of the OC, comprising the lateral sensory and non-sensory domains. A faint expression of Cdh1 could also be seen in a subset of KO cells bordering the medial sensory domain (S3D Fig). Cdh2 was expressed in the medial sensory and non-sensory domains (S3E Fig). The expression of Cdh4 was similar to Cdh2, except its expression was not detected in IPhC (S1F Fig). This superimposition of the combinatorial expression of adhesion molecules and domains in OC suggested an adhesion-based segregation of cells in OC (S3G Fig).

To ask when the putative adhesion code was established, we stained E14.5 for Cdh1, 2, and 4, as well as Sox2 to mark the pro-sensory domain [32]. We find that at E14.5, the expression of Cdh1, 2, and 4 was spatially restricted. Cdh1 shows expression in the lateral regions of the OC, with a faint expression detected on the medial edge of the putative KO domain (Figs 2A and S3H). Cdh2 and Cdh4 are restricted to the medial OC (Fig 2B and 2C). The presence of combinatorial cadherin expression before the onset of differentiation and its persistence during convergent extension-mediated elongation suggested its role in maintaining domain integrity during cochlear morphogenesis (Fig 2D and 2E).

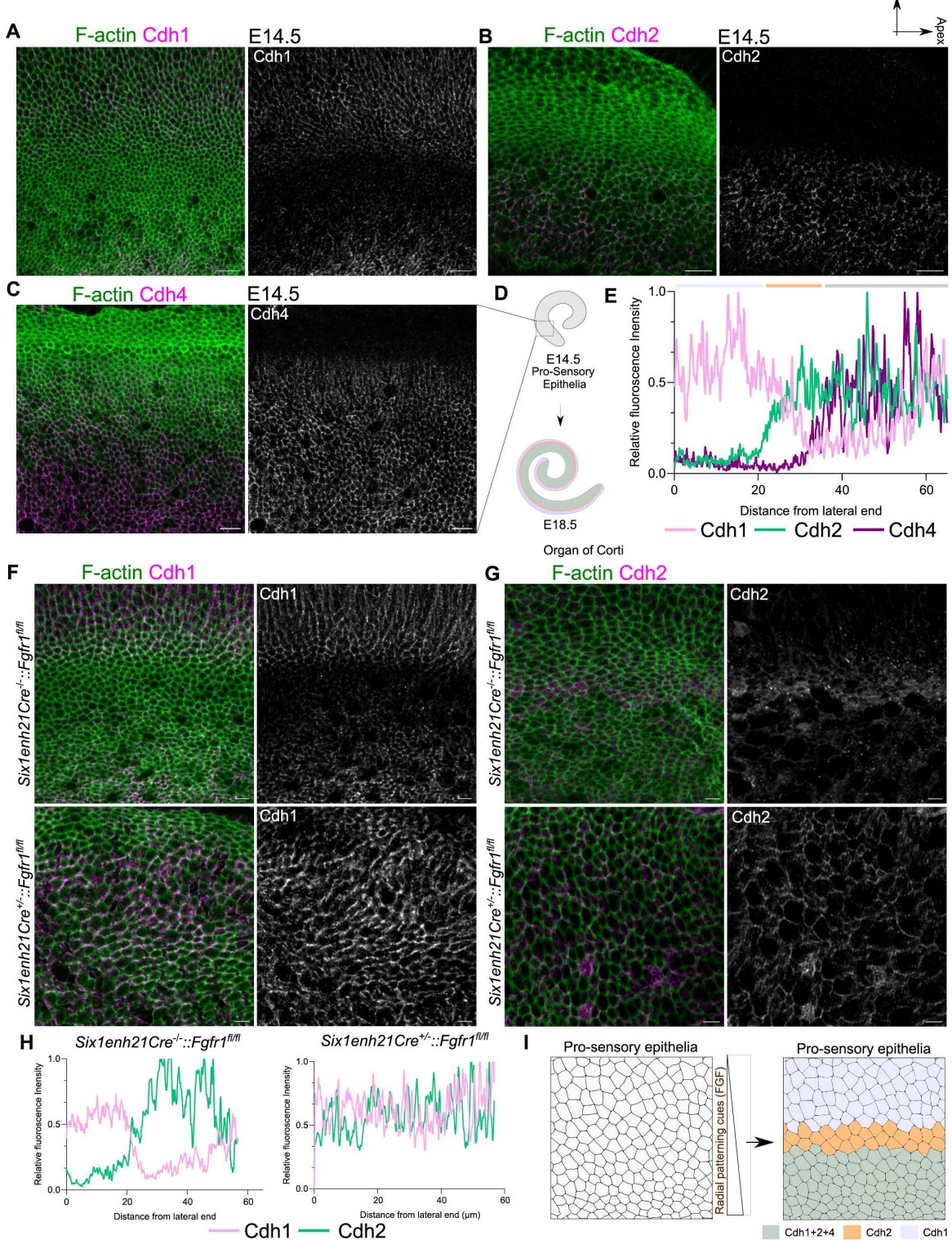

**Fig 2. Radial patterning cues regulate development of adhesion-code-based compartments. (A)** E14.5 OC stained for F-actin (green) and Cdh1 (magenta and gray). **(B)** E14.5 OC stained for F-actin (green) and Cdh2 (magenta and gray). **(C)** E14.5 OC stained for F-actin (green) and Cdh4 (magenta and gray). Image is medially shifted to show the expression in the KO.**(D)** Schematic representing the pro-sensory epithelia at E14.5 develops

into E18.5 OC. **(E)** Relative fluorescence intensity of Cdh1 (pink), Cdh2 (green), and Cdh4 (purple) along the medio-lateral axis of OC at E14.5, representing lateral domain with only Cdh1 (lilac), a Cdh2 expressing domain (yellow), and a medial domain expressing all three cadherins (light green). **(F)** E14.5 OC from control (*Six1enh21Cre*[−/−]*::Fgfr1*[fl/fl]) and fgfr1 mutant (*Six1enh21Cre*[+/−]*:: Fgfr1*[fl/fl]) cochlea stained for F-actin (green) and Cdh1 (magenta and gray). *N* = 3 embryos. **(G)** E14.5 OC from control (*Six1enh21Cre*[−/−]*::Fgfr1*[fl/fl]) and fgfr1 mutant (*Six1enh21Cre*[+/−]*:: Fgfr1*[fl/fl]) cochlea stained for F-actin (green) and Cdh2 (magenta and gray). *N* = 3 embryos.**(H)** Relative fluorescence intensity of Cdh1 (pink), Cdh2 (green) along the medio-lateral axis of OC at E14.5 from control (*Six1enh21Cre*[−/−]*::Fgfr1*[fl/fl]) and mutant (*Six1enh21Cre*[+/−]*:: Fgfr1*[fl/fl]) cochlea. **(I)** Schematic representing the reciprocal expression of cadherins driven by radial patterning cues. Scale Bar: 10 μm. Image orientation: Top is lateral, Right is Apex. Underlying data available in S1 Data.

To test this, we sought to disrupt the differential expression of cadherins. Work on zebrafish neural tube implicated a role for the morphogens involved in cell fate patterning in regulating adhesion molecule expression [29]. In mouse OC, FGF signaling, through Fgfr1, has been shown to influence radial patterning [33,34]. We thus drove the deletion of Fgfr1 from the inner ear using an inner ear-specific Cre (Six1enh21Cre). While E14.5 controls (*Six1enh21Cre*[−/−]*::Fgfr1*[fl/fl]) showed reciprocal expression of Cdh1 and Cdh2 similar to wild type OC; in mutant (*Six1enh21Cre*[+/−]*::Fgfr1*[fl/fl]) cochlea, both Cdh1 and Cdh2 were colocalized to junctions of all cells along the medio-lateral axis of OC, suggesting a loss of the reciprocal expression. (Fig 2F–2I). Immunostaining for Cdh1 and Cdh2 at E18.5 revealed that this misexpression is still evident and correlated with the sporadic presence of HCs in the medial and lateral non-sensory domains (S4 Fig), suggesting a reduction in the domain integrity.

To ask if the control of Cdh1 and Cdh2 expression by FGF signaling is independent of the cell fate specification, we used explant cultures of E16.5 cochlea (S5A Fig). At this stage, cell fates are already established, and CE-mediated elongation is still in progress. E16.5 cochlea explants treated with an inhibitor of Fgfr1, SU5402, for 12 h showed normal expression of p75NTR and BLBP, suggesting negligible impact on cell fate (S5B and S5C Fig). However, staining these Fgfr1-inhibited cochlea for Cdh1 and Cdh2 showed misexpression of Cdh2 (S5D–S5F Fig). Further, Fgfr1-inhibited cochlea, showed ingression of pillar cells, frequent HC–HC contacts, and drifting of HCs into medial and lateral non-sensory domains, suggesting perturbation of domain organization (S6A–S6C Fig).

We next asked whether the cadherin code itself was responsible for preserving domain organization. Using function-blocking antibodies, we investigated whether perturbing cadherin interactions could also perturb OC domains. To first establish that these antibodies were able to recognize Cdh1 and 2 in the cochlea, we cultured E16.5 cochlea explants in the presence of 7D6, a Cdh1 blocking antibody, or 6B3, which blocks Cdh2 interactions, for 1 h and then stained with the respective secondary antibodies [35,36]. We observed 7D6 and 6B3 antibodies localized to the lateral and medial regions of OC, similar to the expression of Cdh1 and Cdh2 (S3D, S3E, S6D, and S6E Figs). We next cultured E16.5 cochlea explants in the presence of either media containing BSA, 6B3 or 7D6 for 12 h. Explants treated with the Cdh2 blocking antibody, 6B3, showed perturbations of the medial sensory and non-sensory domains, with IHCs drifting into the medial non-sensory domain and frequent HC contacts (S6F–S6H Fig). Cochlea explants cultured in the presence of the Cdh1-blocking antibody, 7D6, led to the disruption of organization in the lateral sensory domain, with negligible impact on medial non-sensory and sensory domains (S6F–S6H Fig).

The similarity in SU5402 and 7D6-treated OC and the sustained mis-expression of cadherins in FGF-inhibited cochlea suggests FGF-driven patterned expression of cadherins underlines the domain integrity during convergent extension. Moreover, expression of Cdh1, 2, and 4 was maintained in *Vangl2*[Lp/Lp] mutants that have defective convergent extension, further supporting this inference (S7 Fig).

**Adhesion-code ensures discrete organization of compartments during convergent extension.** Previous work on fly imaginal discs, hindbrain, trachea, ovarian follicles, and somites [37–39] has provided evidence that cells within a compartment undergo distinct cell behaviors. This compartmentalization allows discrete tissue morphogenesis. To ask if the adhesion-code-based compartments in the cochlea also show distinct patterns of organization, we analyzed their development at two scales. The first is to ask how the compartment shape changes, which we refer to as domain

organization. The second is to understand how the organization of cells within each domain changes, referred to as cellular organization. Previous studies have established that the sensory domain elongates (S8A–S8C Fig) along the proximal-distal axis of OC, while HCs and SCs within the domain migrate and intercalate to form a hexagonal lattice-like organization between E15.5 and E18.5 [14,40]. Hence, we focused on the non-sensory domains.

KO is found medial to the sensory domain. Consistent with the increase in cochlear length, the KO also elongates to twice its length along the base-apex axis of the cochlea (2,734±33 µm to 4,702±27 µm). The extension of KO length is accompanied by a decrease in its width at the base and middle turn of the cochlea between E15.5 and E18.5, indicative of convergent extension (Fig 3A and 3B).

To understand the changes in cellular organization, we investigated changes in apical cell shape and size. At E15.5, most cells of the KO have varied apical surface area and a roughly symmetrical aspect ratio (AR=0.87±0.12) close to 1 (Fig 3A and 3C, see Methods for details). However, a small population of cells show an asymmetric elongation, with their long axis aligned perpendicular to the mediolateral axis of the OC (Fig 3D). By E18.5, two populations of cells are visible in the KO. Those at the medial edge of the KO (mKO), have a smaller apical surface area and are elongated, with the long axis parallel to the tissue axis (Fig 3A, 3C, and 3D). At the lateral KO (lKO), closer to the sensory domain, cells have a larger surface area and are elongated orthogonal to the tissue axis (Figs 3A, 3C, 3D and S8). In the lateral non-sensory domain, the apical surface area and the shape index (perimeter/sqrt of area) for both Hensen's and Claudius cells increased between E15.5 and E18.5 (Figs 3E, 3F and S8D, S8E).

Our data suggest that the non-sensory domains of OC, similar to the sensory domains, undergo distinct and discrete reorganization during cochlear morphogenesis, a hallmark of compartment behavior. To ask if disrupting compartments also leads to a disruption in both domain and cellular organization of the non-sensory compartments, we used mutants in which the adhesion code, and thus compartment cohesion, was disrupted. In *Fgfr1* mutants, we observed a reduction in the difference between the apical surface area of the cells of mKO and lKO (S8F and S8H Fig). The cells from mutants also showed a reduction in circularity (S8F and S8I Fig). Further, the Hensen's cells from mutants showed an increase in the shape index with similar apical surface area (S8J–S8L Fig), suggesting a decrease in the compartment-specific organization.

**PCP regulated NMII-activity drives distinct cellular organization.** To understand the mechanisms behind discrete cellular organization in compartments of the OC, we first looked at proliferation. We injected EdU into pregnant females at E13.5, E15.5, E16.5, and E18.5 and fixed embryos 6 h post-injection (S9A Fig). At E13.5 (+6 h), we observed EdU was incorporated into the entire OC (S9B Fig), similar to previous observations [18]. By E15.5, the number of EdU-positive cells at the base decreased to less than 10%. By E16.5, proliferation had ceased and remained so till E18.5 (S9B Fig). As the KO increased in length between E15.5 and E18.5, we concluded that there is a limited contribution by proliferation. We next asked if cellular rearrangements could contribute to compartment reorganization. In epithelia, cellular rearrangements result from neighbor exchange, with an obligatory intermediate step where 4 or more cells meet at a vertex. We thus assessed the number of 4-cell vertices between E15.5 and E18.5. At E15.5, 33% of all vertices in the sensory compartment have 4 or more cells. This decreases significantly such that by E18.5 only 20% of vertices are made up of 4 or more cells (S9D Fig).

Similarly, in the KO domain, mKO showed 40% of vertices to have 4 or more cells, suggesting a higher rate of cellular reorganization. In the lKO, only 20% of the vertices showed 4 or more cells, suggesting a lower rate of reorganization compared to the mKO. The proportion of vertices with 4 or more cells decreased in the lKO by E18.5. The proportion of vertices with 4 or more cells in the mKO at E18.5 remained equivalent to the numbers observed at E15.5 (S9E Fig). This data suggests that cells in the KO domain undergo cell rearrangement, higher at the medial edge compared to the lateral edge. In the absence of cell division, we hypothesized that this cellular rearrangement drives cochlear morphogenesis.

To test this, we sought to perturb the process of cellular reorganization by disrupting the activity of the acto-myosin complex, essential for cellular intercalations. Non-muscle myosin (NM) forms the motor component of the acto-myosin complex. The motor activity of NMII is regulated by the phosphorylation status of its regulatory light chain (RLC). Thus,

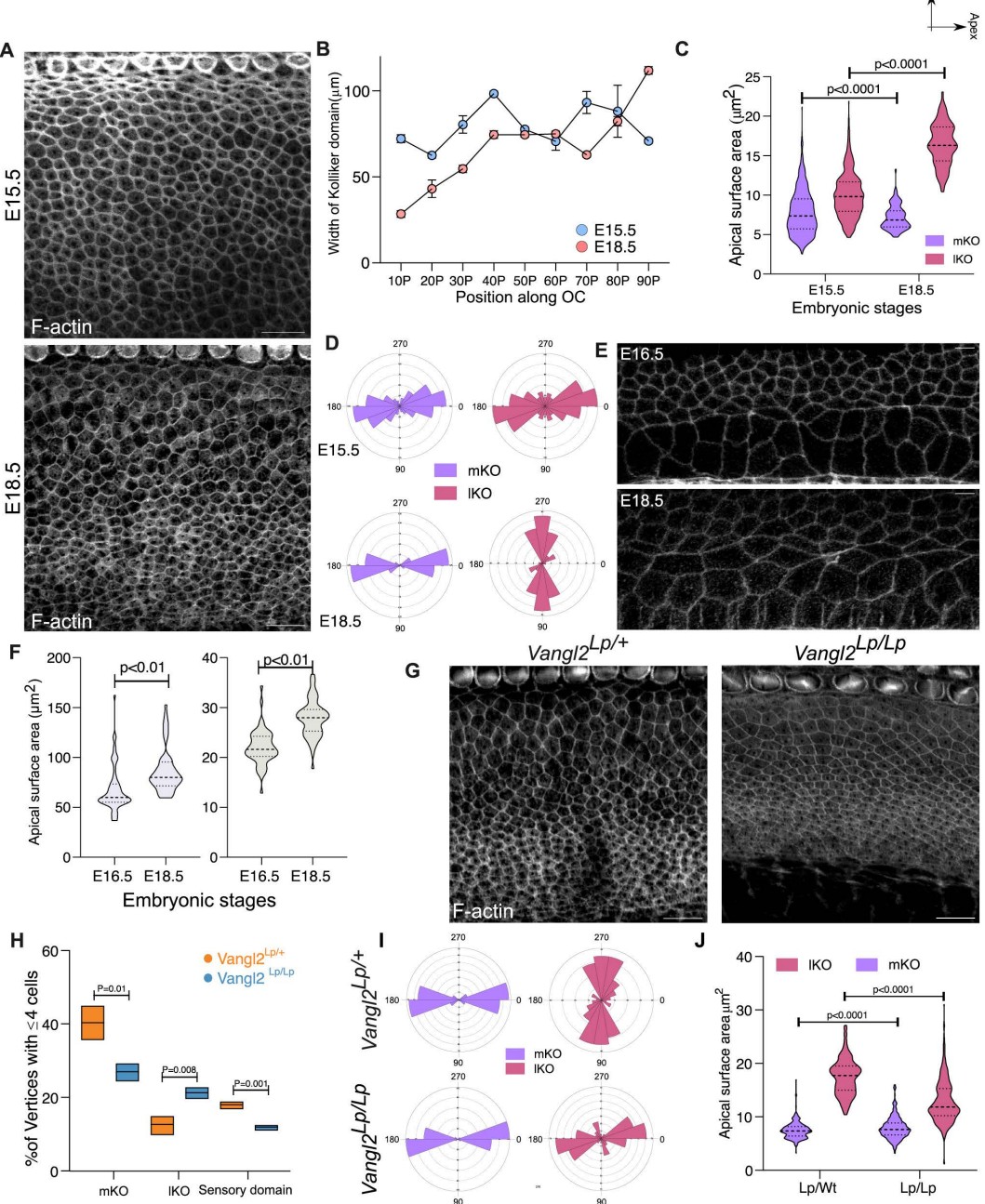

**Fig 3. Planar cell polarity cues regulate organization of each domain. (A)** Base region (30P) of OC from E15.5 and E18.5 stained for F-actin (gray) representing the KO domain. **(B)** Width of the Kölliker's organ along the OC at E15.5 and E18.5. $N = 4$ cochlea for each stage. **(C)** Apical surface area of mKO and lKO at E15.5 and E18.5 at basal region (30P). $N = 378/303$ (mKO/lKO, E15.5) and 283/298 (mKO/lKO, E18.5); 4 cochlea each stage. **(D)** Rose stack plot represents the elongation cell axis at mKO and lKO at E15.5 and E18.5. $N = 378/303$ (mKO/lKO, E15.5) and 283/298 (mKO/lKO, E18.5); 4 cochlea each stage. **(E)** Base region (30P) of OC from E15.5 and E18.5 stained for F-actin (gray) representing the lateral non-sensory domain containing Hensen's cell and Claudius cells. **(F)** Apical surface area of Hensen's and Claudius cells at base region of OC at E16.5 and E18.5. $N = 64/60$ for Hensen and 57/59 for Claudius (E16.5/E18.5). **(G)** Base region (30P) of E18.5 OC from heterozygous (*Vangl2* $^{Lp/+}$) and homozygous (*Vangl2* $^{Lp/Lp}$) *looptail* mutant stained for F-actin (gray). **(H)** Percentage of vertices with 4 or more cells for mKO, lKO, and sensory domain in OC from heterozygous (*Vangl2* $^{Lp/+}$) and homozygous (*Vangl2* $^{Lp/Lp}$) *looptail* mutant at E18.5. $N = 3$ cochlea each: 250/622 mKO; 113/923 lKO; 155/880 sensory in Het and 249/921 mKO; 261/1215 lKO; 131/1079 sensory in Homo. (4 or more cell vertices/Total vertices). **(I)** Rose stack plot representing the axis of cell elongation for mKO and lKO cells at the base region (30P) of E18.5 OC from heterozygous (*Vangl2* $^{Lp/+}$) and homozygous

(*Vangl2 ^Lp/Lp*) looptail mutant. *N* = 218/224 (mKO/lKO, Het) and 234/198 (mKO/lKO, Homo). **(J)** Apical surface area of mKO and lKO cells at Base region (30P) of E18.5 OC from heterozygous (*Vangl2 ^Lp/+*) and homozygous (*Vangl2 ^Lp/Lp*) looptail mutant. *N* = 218/224 (mKO/lKO, Het) and 234/198 (mKO/lKO, Homo). Scale Bar: 10 µm. Unpaired *T* test, Image orientation: Top is lateral, Right is Apex. Underlying data available in S1 Data.

we first immunostained OC for mono and di-phosphorylated forms of RLC. At E18.5, p-RLC is expressed on all junctions (S10A Fig). However, the pp-RLC is localized at the medial edge of OHC-DC junctions (S10B and S10B′ Fig) and IPhC-Pillar cells junctions. In the KO compartment, pp-RLC was localized along the junctions of the long axis of KO cells (S10B Fig). To test their role in morphogenesis of the cochlea, we used our ex vivo explant method to culture E16.5 cochlea for 8 h, in the presence or absence of Myosin Light Chain Kinase inhibitor, ML7 (which inhibits RLC phosphorylation) [41,42]. In MLCK-inhibited OC, we observed a decrease in apical surface area and circularity of OHC compared to the control samples (S10C–S10H Fig), suggesting a decrease in spatial organization within the sensory domain. Previous work on avian auditory epithelia has shown that the spatial organization of HC is coupled to the alignment of HC polarity to the tissue axis [12]. Similarly, we observed a decrease in the alignment of HC polarity for IHC and OHC in the MLCK-inhibited cochlea (S10E and S10F Fig). In addition, the difference in the apical surface area and the elongation axis of mKO and lKO cells was also reduced in the MLCK-inhibited OC (S10I–S10K Fig). This data suggested that NMII-driven neighbor-exchange drives reorganization of sensory and non-sensory domains during development.

To further understand this organization, we decided to understand how NMII activity is regulated in each compartment. Previous studies, including our work on avian auditory epithelia, have shown that PCP cues through Vangl2 regulate RLC phosphorylation [12,43–45]. In mouse, the expression of Vangl2 largely overlapped with the expression of pp-RLC (S11A and S11B Fig). Hence, we used *Vangl2^Lp/Lp* mutant, which, as previously reported, shows a reduction in alignment of HC polarity (S11C and S11D Fig) [16,22]. pp-RLC shows a down-regulation on the junctions of both sensory and non-sensory compartments in *Vangl2^Lp/Lp* mutants, while the p-RLC was comparable to the littermate controls (S11E–S11H Fig). At the scale of the domain, the *Vangl2^Lp/Lp* mutants showed a decrease in the width of both the KO and sensory domains at the base and middle turn of OC (Figs 3G and S11I–S11J). Further, at E18.5, *Vangl2^Lp/Lp* mutants also showed a significant decrease in the number of 4 cell vertices in both mKO and sensory domain, suggesting a decrease in cell intercalation (Fig 3H). Interestingly, mutants showed a significant increase of four cell vertices in the lKO. The differences in the apical surface area and the preferential axis of elongation for mKO and lKO cells were also reduced in the mutants (Fig 3G, 3I, and 3J). The regulation of RLC phosphorylation by Vangl2 and the similarity of *Vangl2^Lp/Lp* mutants with MLCK inhibited OC, suggests that Vangl2-regulated NMII activity may govern the organization of not only the sensory compartments of the OC but also the non-sensory compartments.

**Compartment-intrinsic remodeling has extrinsic effects on cellular organization.** To understand the effect of disrupting cell and domain organization in a compartment, we used the Cre-LoxP system to delete Vangl2 from defined domains. We generated mice carrying one copy of the *looptail* mutation of Vangl2 and the other allele, the conditional Vangl2 mutant, where the coding sequence is flanked by loxP sites. We referred to this mouse as *Vangl2^lp/fl*.

Previous studies using an inducible CreER line driven by Neurogenin, identified expression in cells within the KO, as well as in the cochleovestibular ganglion. Importantly, cells within the sensory compartment of the OC were not labeled [46]. We first verified that a non-inducible Cre-line, *Ngn1^457-Cre* could recapitulate this expression domain [47]. *Ngn1^457-Cre* males were crossed with *Ai14* (ROSA26-TdTomato) females to generate *Ngn1^457-Cre::Ai14*. OC from E18.5 embryos were dissected and imaged. Similar to Ngn1-CreER, we found that *Ngn1^457-Cre* could drive recombination in around 10% of cells located throughout the KO (Fig 4A and 4B). Recombination was not detected in the sensory compartment. At E18.5, the cochlea from the experimental animals (*Ngn1^457-Cre^+/−:: Vangl2^lp/fl*) showed a reduction in difference of apical surface area between mKO and lKO cells compared to the littermate control ((*Ngn1^457-Cre^−/−:: Vangl2^lp/fl*)) cochlea (Fig 4C and 4D). Additionally, in (*Ngn1^457-Cre^+/−:: Vangl2^lp/fl*) mutants the long axis of lKO cells was alligned parallel to the OC

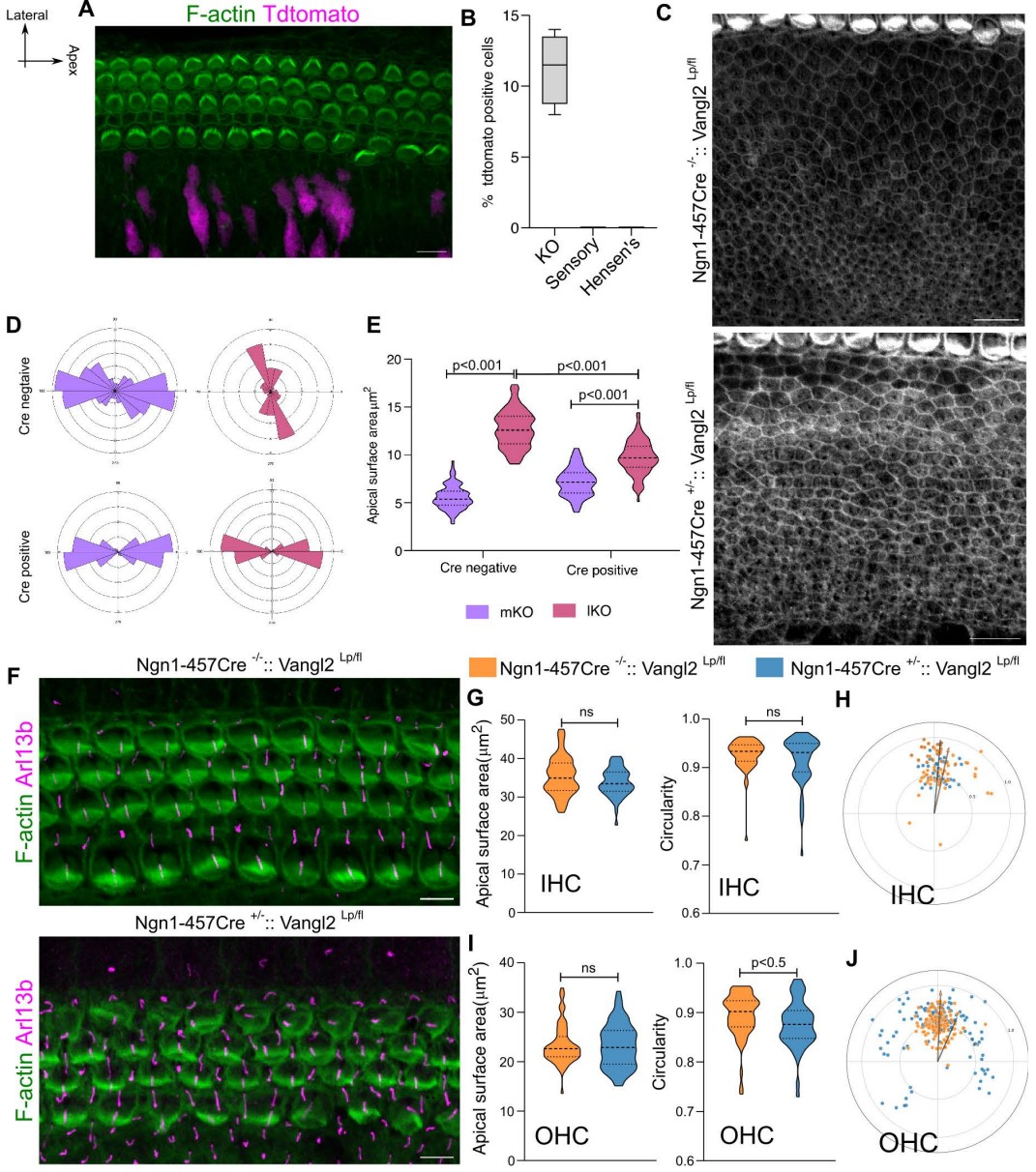

**Fig 4. Deletion of Vangl2 from Kölliker's organ disrupts organization in lateral non-sensory domain. (A)** E18.5 OC from *Ngn1⁴⁵⁷-Cre⁺ᐟ⁻::Ai14*, stained with F-actin (green) showing cre-mediated expression of Tdtomato (magenta) in Kölliker's organ and not in sensory domain. **(B)** Percentage of tdtomato positive cells from the *Ngn1⁴⁵⁷-Cre⁺ᐟ⁻::Ai14* cochlea in KO, sensory and lateral non-sensory domain. *N*=3. **(C)** Kölliker's organ from base region (30P) of OC from Cre negative control (*Ngn1⁴⁵⁷-Cre⁻ᐟ⁻:: Vangl2ᴸᵖ/ᶠˡ*) and Cre positive mutants (*Ngn1⁴⁵⁷-Cre⁺ᐟ⁻:: Vangl2ᴸᵖ/ᶠˡ*) stained for F-actin (gray). *N*=4 embryos. **(D)** Rose stack plots representing the axis of cell elongation in medial and lateral KO cells from Cre negative control (*Ngn1⁴⁵⁷-Cre⁻ᐟ⁻:: Vangl2ᴸᵖ/ᶠˡ*) and Cre positive mutant (*Ngn1⁴⁵⁷-Cre⁺ᐟ⁻:: Vangl2ᴸᵖ/ᶠˡ*) OC. *N*=190/202 for Control and 198/208 for Mutant (mKO/lKO). **(E)** Apical surface area of medial and lateral KO cells from Cre negative control (*Ngn1⁴⁵⁷-Cre⁻ᐟ⁻:: Vangl2ᴸᵖ/ᶠˡ*) and Cre positive mutant (*Ngn1⁴⁵⁷-Cre⁺ᐟ⁻:: Vangl2ᴸᵖ/ᶠˡ*) OC. *N*=190/202 for Control and 198/208 for Mutant (mKO/lKO). **(F)** Base region (30P) of OC from Cre negative control (*Ngn1⁴⁵⁷-Cre⁻ᐟ⁻:: Vangl2ᴸᵖ/ᶠˡ*) and Cre positive mutant (*Ngn1⁴⁵⁷-Cre⁺ᐟ⁻:: Vangl2ᴸᵖ/ᶠˡ*) stained for F-actin (green) and Arl13b (magenta). **(G)** Apical surface area and Circularity of IHC of Cre negative control (*Ngn1⁴⁵⁷-Cre⁻ᐟ⁻:: Vangl2ᴸᵖ/ᶠˡ*) in orange and Cre positive mutant (*Ngn1⁴⁵⁷-Cre⁺ᐟ⁻:: Vangl2ᴸᵖ/ᶠˡ*) in blue. *N*=92/96 (control/mutant). **(H)** Polar coordinates representing the position of kinocilia from IHC of Cre negative control (*Ngn1⁴⁵⁷-Cre⁻ᐟ⁻:: Vangl2ᴸᵖ/ᶠˡ*) in orange and Cre positive mutant (*Ngn1⁴⁵⁷-Cre⁺ᐟ⁻:: Vangl2ᴸᵖ/ᶠˡ*) in blue. *N*=66/66 (control/mutant). **(I)** Apical surface area and Circularity of OHC of Cre negative control (*Ngn1⁴⁵⁷-Cre⁻ᐟ⁻:: Vangl2ᴸᵖ/ᶠˡ*) in orange and Cre positive mutant (*Ngn1⁴⁵⁷-Cre⁺ᐟ⁻:: Vangl2ᴸᵖ/ᶠˡ*) in blue. *N*=143/151 (control/mutant). **(J)** Polar coordinates representing the position of kinocilia from OHC of Cre negative control (*Ngn1⁴⁵⁷-Cre⁻ᐟ⁻:: Vangl2ᴸᵖ/ᶠˡ*) in orange and Cre positive mutant (*Ngn1⁴⁵⁷-Cre⁺ᐟ⁻:: Vangl2ᴸᵖ/ᶠˡ*) in blue. *N*=113/121 (control/mutant). Scale Bar: 10 μm in **A**, **C** and 5 μm in **F**. Unpaired *T* test, ns=*P*>0.05, non-significant. Image orientation: Top is lateral, Right is Apex. Underlying data available in S1 Data.

tissue axis, whereas in controls, the axis was orthogonal (Fig 4C). This suggests that compartment-specific deletion of Vangl2 perturbs local cellular organization.

To test if this local perturbation of KO organization had non-local effects, we examined the adjacent sensory domain, which comprises of the medial and lateral sensory compartments. The immediately adjacent medial sensory compartment showed no significant defects in polarity, apical surface area or circularity (Fig 4F–4H). However, in the lateral sensory compartment of *Ngn1⁴⁵⁷-Cre⁺/⁻:: Vangl2ˡᵖ/ᶠˡ* mutants, OHC3 cells displayed reduced planar polarity alignment with OHC showing reduced circularity, although surface area was unaffected (Fig 4F, 4I, and 4J). This suggested that disruption of *Vangl2* from the non-sensory KO compartment could influence the development of planar polarity in the lateral sensory compartment.

Previous work on the vestibular sensory epithelium [48], wing imaginal disc [49], and ommatidia [50], along with modeling studies [51,52], describe a phenomenon known as domineering non-cell autonomy. Here, a local loss of PCP influences neighboring cells, with effects that diminish with distance. However, in *Ngn1⁴⁵⁷-Cre⁺/⁻:: Vangl2ˡᵖ/ᶠˡ* mutants, only the polarity of OHC3 cells is affected (Fig 4J). These are at a considerable distance from the KO. As polarity in IHC of the medial sensory compartment is more developmentally advanced than OHC [53], it is possible that any defects in IHC polarity could have been observed at an earlier developmental stage. However, even at E16.5, IHC showed no disruption in the alignment of planar polarity. In contrast, the 3rd row of OHC of the lateral sensory compartment still showed a small deviation in HC polarity alignment as compared to the littermate control (S12A Fig). Our *Ngn1⁴⁵⁷-Cre* driven deletion of Vangl2 from the KO domain also contains a *looptail* allele, which has been suggested to exert a dominant phenotype. Although the comparison with the *Ngn1⁴⁵⁷-Cre⁻/⁻:: Vangl2ˡᵖ/ᶠˡ* littermate controls suggest that the *looptail* allele does not show dominance, we assayed the phenotype of the *Vangl2ᶠˡ/ᶠˡ* mutant. Using the *Ngn1⁴⁵⁷-Cre⁺/⁻:: Vangl2ᶠˡ/ᶠˡ*, where both alleles are floxed out by Cre-mediated recombination we find, similar to *Ngn1⁴⁵⁷-Cre⁺/⁻:: Vangl2ˡᵖ/ᶠˡ,* that *Ngn1⁴⁵⁷-Cre⁺/⁻:: Vangl2ᶠˡ/ᶠˡ* also showed a perturbation of the polarity of 3rd row of OHC (S12B Fig). This suggests that the loss of OHC3 alignment results from the loss of cellular organization in the KO compartment. This can neither be explained by domineering non-cell autonomy nor a dominant effect of *looptail* mutation. Instead, we suspected a long-range effect between compartments of the OC.

To assess a reciprocal influence of the sensory domain on the KO compartment, we used *Lgr5-CreERT2*, a tamoxifen-inducible Cre which, when induced at E15.5, drives recombination in the sensory domain but not in the KO (S12C and S12D Fig). In the animals where recombination is induced at E15.5, we observed a reduction in planar polarity and circularity of HCs at E18.5 (Fig 5A–5E). Analysis of the KO compartment revealed a decreased difference in the apical surface area between the mKO and lKO cells and a loss in the normal mediolateral-orthogonal axis difference (Fig 5F–5H). This data indicates a reciprocal, non-local influence on KO compartment organization (Fig 5I).

Our results show that deletion of *Vangl2* from a single compartment affects cellular organization across distant domains of the OC. This bidirectional interaction between compartments suggests a mechanism of compartment coupling, which coordinates morphogenetic organization across spatially distinct epithelial domains.

**Mechanical origin of coupling.** There are potentially multiple mechanisms that could mediate coupling among compartments. Given that we observed coupling in domain-specific mutants of *Vangl2*, and considering that Vangl2 regulates junctional mechanics, we hypothesized that this coupling might involve a mechanical component. Previous studies have shown that Vinculin, a protein critical for transmitting force at cell junctions [54], is expressed in the sensory domain and is regulated by PCP cues [55]. To explore this possibility, we immunostained the OC for Vinculin.

At E15.5, Vinculin localized broadly to all junctions within the OC (S13A Fig). By E18.5, its expression became more spatially patterned, with significantly higher mean intensity on junctions in Hensen's cells, and lKO cells compared to other sensory and non-sensory cell types (S13B Fig). Notably, this pattern of differential Vinculin localization was reduced in *Vangl2ᴸᵖ/ᴸᵖ* mutants (S13C and S13D Fig), consistent with the idea that Vinculin enrichment may be regulated by PCP activity.

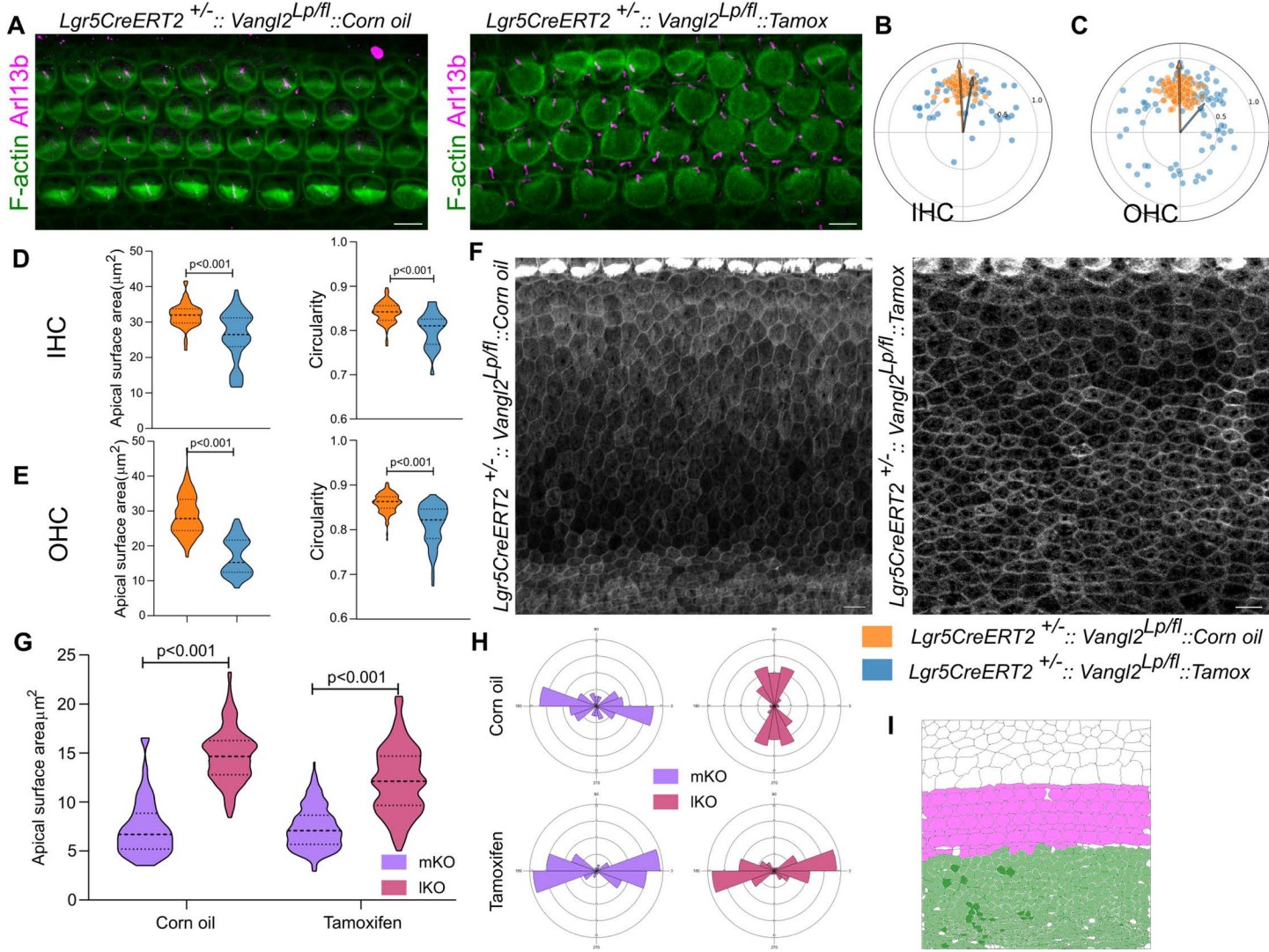

**Fig 5. Deletion of Vangl2 from sensory compartments disrupts organization in KO compartment. (A)** Base region (30P) of OC from corn oil injected control (*Lgr5CreERT2+/−:: Vangl2lp/fl::Corn oil*) and tamoxifen induced mutant (*Lgr5CreERT2+/−:: Vangl2lp/fl::Tamox*) stained for F-actin (green) and Arl13b (magenta). **(B)** Polar coordinates representing the position of kinocilia of IHC from corn oil injected control OC in orange and tamoxifen-induced mutant in blue. $N = 66/66$ (Corn oil/Tamoxifen). **(C)** Polar coordinates representing the position of kinocilia of OHC from corn oil injected control OC in orange and tamoxifen-induced mutant in blue. $N = 110/110$ (Corn oil/Tamoxifen). **(D)** Apical surface area and Circularity of IHC from corn oil injected control OC in orange and tamoxifen-induced mutant in blue. $N = 89/99$ (corn oil/Tamoxifen). **(E)** Apical surface area and circularity of OHC from corn oil injected control OC in orange and tamoxifen-induced mutant in blue. $N = 109/128$ (Corn oil/Tamoxifen). **(F)** Kölliker's organ from base region (30P) of OC from corn oil-injected control OC and tamoxifen-induced mutant embryos stained for F-actin(gray). Scale Bar: 10 μm. **(G)** Apical surface area of medial and lateral KO cells from corn oil injected control and tamoxifen-induced mutant OC. $N = 265/245$ for Corn oil and 264/234 for Tamoxifen (mKO/lKO). **(H)** Rose stack plots representing the axis of cell elongation in medial and lateral KO cells from corn oil injected control and tamoxifen induced mutant OC. $N = 265/245$ for Corn oil and 264/234 for Tamoxifen (mKO/lKO). **(I)** Schematic representing the expression of cre in lateral and medial sensory compartments (magenta) and hence the deletion of Vangl2, with the effect on the organization in the KO (green). Scale Bar: 5 μm in A and 10 μm in **F**. Unpaired *T* test. Image orientation: Top is lateral, Right is Apex. Underlying data available in S1 Data.

These findings led us to hypothesize that Vinculin may contribute to mechanical coupling, potentially acting as a mediator of inter-compartmental force transmission. To test this, we crossed a conditional allele of Vinculin (*Vincfl/fl*) with *Emx2-Cre*, deleting vinculin from the E12.5 cochlea. We could only obtain 3 experimental embryos out of 114 examined

(see method for details). In these mutants, we observed a disruption in the alignment of planar polarity in the lateral sensory compartment (Fig 6B–6F). The overall width of the KO compartment in mutants remained comparable to controls, however we noted a subset of lKO cells, adjacent to the medial sensory compartment that showed a larger surface area (Figs 6B, 6G, 6H, and S13E), an effect opposite to that observed in *Vangl2^{Lp/Lp}* mutants (Fig 3J). Moreover, the differences in the direction of the long axis between lKO and mKO cells were greatly reduced in *Emx2-Cre^{+/−}::Vinc^{fl/fl}* (Fig 6G and 6I).

Together, these results demonstrate that loss of Vinculin disrupts cellular organization in both sensory and non-sensory compartments. These findings support a model where coupling between compartments is, at least in part, mediated through vinculin-dependent junctional mechanics. This disruption in the organization of cells in both sensory and non-sensory compartments suggested that coupling between compartments is mediated, at least in part, through vinculin-dependent junctional mechanics.

## Discussion

Using the mouse auditory epithelia, we investigate a fundamental question in morphogenesis: How are developmental instructions coordinated across spatial scales so that tissues form shapes and patterns that are robust and reproducible? We demonstrate that the OC is segregated into adhesion-code-based spatially restricted developmental units called compartments (Fig 7A). These compartments allow for locally coordinated, discrete cell behaviors driven by cell-intercalation, shape changes, and growth. Using compartment-specific perturbations of cell mechanics, we find intra-compartment defects have effects on the cellular organization of other compartments. We consider this as evidence of inter-compartment coupling and suggest that it provides a mechanism allowing the transmission of morphogenetic information across spatial hierarchies, ensuring fidelity in organization (Fig 7A).

Compartment coupling has been proposed as a mechanism to coordinate adjacent domains. This is the case during axial elongation [56], where force coupling ensures the coordination of neighboring axial and paraxial mesoderm. Similarly, recent studies during hair follicle development have found coupling between adjacent compartments necessary for the invagination and polarity of the feather bud and hair follicle [57–59]. The behaviors between adjacent compartments can be considered linear, and studies from the coordination of the prechordal plate with the anterior neural plate in the zebrafish suggest that cells further from the compartment interface show a reduced coupling effect [60]. In contrast, the coupling shown in the mouse OC is both linear, as evident by disorganization of KO compartment in Lgr5-driven cKO from sensory domain, and non-linear, in misalignment of OHC3 and not in intervening HCs in Ngn1-457 driven cKO from KO compartment. This non-linear influence points to a more complex, potentially hierarchical form of mechanical communication between compartments that not only utilizes local force transmission but also integrates signals over longer distances through emergent properties of tissue mechanics.

Work on spiraling of the cochlea emphasizes the importance of coupling in the cochlea [17,18]. The bending of the cochlea emerges from two types of cellular dynamics. The first is localized to the medial non-sensory domain and involves nuclear stalling at the luminal side of the duct, which leads to a bending of the cochlear duct [17]. The second is a cell flow directed by helical ERK waves originating at the tip of the cochlear duct, which directs a reciprocal lateral to medial cell flow [18]. Importantly, this coupling is bidirectional: medial bending due to luminal nuclear stalling generates stress that must be dissipated laterally, while lateral flows driven by ERK dynamics may reciprocally constrain medial curvature and polarity. In addition, shear forces generated by the lateral Hensen's cells have been proposed to refine the cellular pattern in the lateral sensory domain, inducing hexatic ordering in this compartment [11]. This again emphasizes the need to reinforce adhesion-based compartments. These findings support a model in which differential growth between medial and lateral domains generates geometric constraints that are transmitted across compartments, necessitating mechanical coupling to maintain pattern fidelity and planar polarity. Disruptions in any of these components, such as adhesion, contractility, or directional signaling, are sufficient to impair cellular alignment and polarity [9,12,40]. These phenotypes underscore the interdependence of signaling, mechanical force, and morphogenetic patterning across compartments.

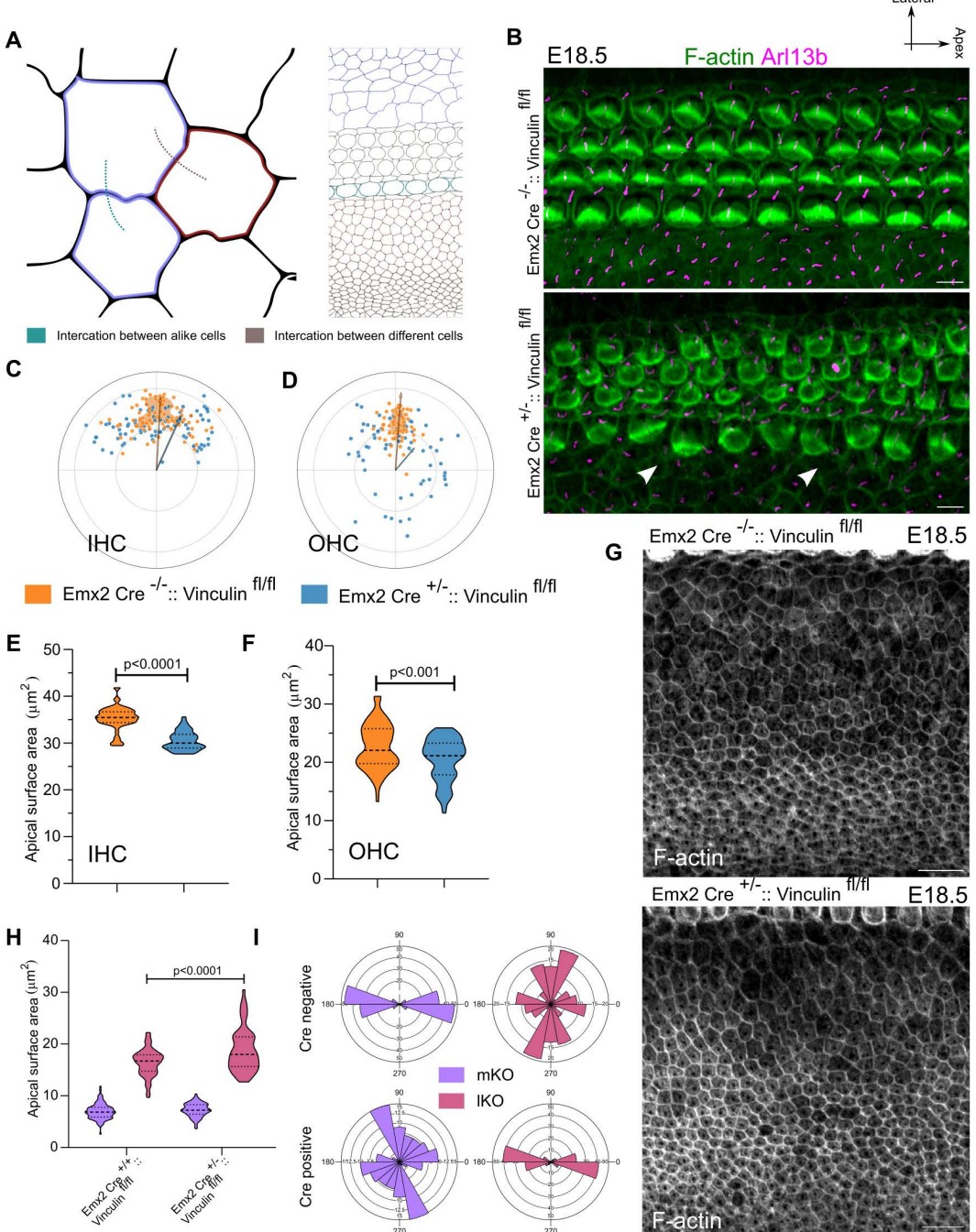

**Fig 6. Disruption of Junctional force transmission complex protein, Vinculin, disrupts the organization of OC. (A)** Schematic representing the communication between cells of OC with diverse mechanical properties. We consider the cells of OC are mechanically different and the communication can be between cells of same type and cells of different types. **(B)** Base region (30P) of E18.5 OC from the control (*Emx2-Cre⁻/⁻:: Vinculin fl/fl*) and mutant for vinculin (*Emx2-Cre⁺/⁻:: Vinculin fl/fl*) stained for f-actin (green) and Arl13b (magenta). Arrow represents the disrupted organization of Inner border cells. **(C)** Polar coordinates representing the Kinocilium position of IHC from control (orange) and vinculin mutant (blue) at base region (30P) of OC. *N* = 66/66 (control/Mutant). **(D)** Polar coordinates representing the Kinocilium position of OHC from control (orange) and vinculin mutant (blue) at base region (30P) of OC. *N* = 68/68 (Control/Mutant). **(E)** Apical surface area of IHC from control (orange) and vinculin mutant (blue) at base region (30P) of OC. *N* = 66/66 (Control/Mutant). **(F)** Apical surface area of IHC from control (orange) and vinculin mutant (blue) at base region (30P) of OC. *N* = 66/66 (Control/Mutant). **(G)** KO of base region of OC (30P) from control and Vinculin mutant OC stained for F-actin (gray). **(H)** Apical surface area of medial and lateral KO cells

from base region (30P) of control and Vinculin mutant OC. *N* = 145/165 (Control/Mutant). **(I)** Rose stack plots representing the axis of cell elongation in medial and lateral KO cells from base region (30P) of control and Vinculin mutant OC. *N* = 145/165 (Control/Mutant). Scale Bar: 5 μm in B and 10 μm in **G.** Unpaired *T* test. Image orientation: Top is lateral, Right is Apex. Underlying data available in S1 Data.

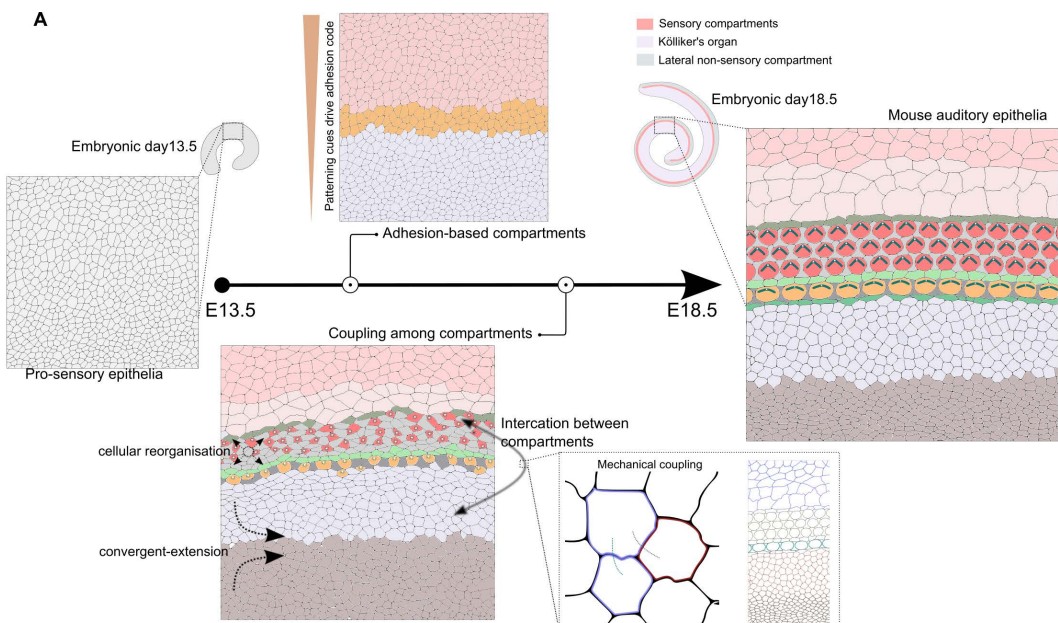

**Fig 7. Model showing the adhesion code-based compartmentalization and compartment coupling governs pattern formation in inner ear. (A)** The developing organ of Corti undergoes FGF signaling-driven compartmentalization into adhesion-code-based domains. This domain organization allows each domain to undergo a discrete cellular organization. The organization in one domain have linear and non-linear influences on the organsiation of another domain, called compartment coupling. Compartment coupling has a mechanical origin.

Our findings propose compartments as fundamental morphogenetic modules that enable the emergence of distinct cellular organization within a shared epithelial field. By defining domains with unique adhesion codes, mechanical properties, and signaling states, compartments allow for the independent regulation of cell behaviors within a contiguous tissue. At the same time, our data reveal that coupling between compartments is essential for coordinating these behaviors across the organ, thus ensuring overall structural and functional coherence. Importantly, we propose that compartments serve as intermediaries bridging the scale of embryonic fields and individual cellular interactions. Through mechanochemical coupling mechanisms, compartments transmit and integrate positional information, enabling robust patterning despite local perturbations. This coupling is not static: it appears to depend on the degree of intra-compartmental organization, suggesting that the competence for coupling is temporally regulated. In this way, compartment coupling provides not just spatial coordination but also introduces a temporal logic to morphogenetic progression. Taken together, we propose that compartments and their coupling constitute a generalizable framework for organizing morphogenetic information in both space and time. While this concept is exemplified in the cochlea, with its precisely arranged HCs and complex spiraling morphogenesis, we anticipate that similar principles apply broadly across organ systems where patterned growth, tissue polarity, and spatial integration are required.

## Methods

### Mice

**Animal housing.**  All mice were housed at Animal Care and Resource Centre (ACRC) at NCBS, in accordance with Institutional Animal Ethics Committee guidelines. The experiments on mouse were approved by Institutional Animal Ethics Committee with approval number NCBS-IAE-2020/13(R1M_EE).

**Mouse strains.**  Mouse strains used are presented in S1 Table of Supporting information

**Genotyping.**  For maintaining stock animals of strains, an ear biopsy was collected from P21 to P30 animals. Each biopsy was lysed in 150 µl lysis solution (250 µl of 1M NaOH, 2 µl of 1M EDTA in 9.748 ml of autoclaved water) for 1 h at 95 °C in a 1.5 ml microcentrifuge tube and then stored after adding 150 µl neutralizing solution (0.4 ml of 1M Tris-Cl in 9.6 ml of autoclaved water). 1 µl of this solution was used as template for polymerase chain reaction (PCR) based genotyping. A 10 µl of PCR mixture contained 5 µl of Kappa-2× genotyping master mix (Cat# KK1024, Roche), 0.5 µl each of reverse and forward primer, and 3 µl of nuclease-free water. The PCR was performed as per manufacturer protocol and the oligonucleotide used as primers is listed in S2 Table of Supporting information.

*Looptail* mutants were identified with kinked to looped tails in heterozygous conditions and with an open neural tube in the homozygous conditions.

**Note:** The Vinculin flox mouse was crossed with both Sox10-Cre and Emx2-Cre to obtain Vinculin cKOs for analysis. However, we could not find any embryos at E18.5 for *Sox10-Cre$^{+/-}$::Vinc$^{fl/fl}$*. Only 3 out of 114 *Emx2-Cre$^{+/-}$::Vinc$^{fl/fl}$* animals were obtained. The heterozygous *Emx2-Cre$^{+/-}$::Vinc$^{+/fl}$* developed a hydrocephalous-like phenotype.

**Tamoxifen Injection.**  The age-appropriate female of the *Vangl2$^{fl/fl}$* genotype was bred with *Lgr5-CreERT2$^{+/-}$:: Vangl2$^{Lp/fl}$* genotype males. Tamoxifen was dissolved in 90% corn oil and 10% ethanol to prepare a stock concentration of 10 mg/ml. Pregnant females were weighed at the E15.5 stage using an electronic balance and were injected intraperitoneally with an appropriate volume of tamoxifen solution at 4.5 mg/40 g of body mass. As a control, corn oil and ethanol solution without tamoxifen is injected.

**EdU injection.**  The age-appropriate female were injected with EdU (2.5 mg/40g of animal weight) intraperitoneally using sterilized syringe and were dissected after 6 h.

### Immunostaining

The inner ears from staged mouse embryos were dissected using a pair of forceps in ice-cold Phosphate-Buffered Saline (PBS) and then fixed in 4% Paraformaldehyde (PFA) for 30 min to overnight according to the primary antibody, see S3 Table of Supporting information. The inner ear is further dissected to expose sensory epithelia. The sensory epithelia are then permeabilized using 0.3% Tween-20 in PBS for 30 min at room temperature and blocked using blocking solution (5% heat-inactivated goat serum, 1% bovine serum albumin, 0.3% Tween-20 and PBS) for 1 h at room temperature. Sensory epithelia are then incubated with primary antibodies diluted in a blocking solution (5% heat-inactivated goat serum, 1% Bovine Serum Albumin, 0.3% Tween-20 in PBS) for overnight at 4 °C. Sensory epithelia are then washed for 4 h with a change of washing solution (0.3% Tween-20 in PBS) after every 30 min. The sensory epithelia are then incubated with secondary antibodies conjugated with Alexa fluor and phalloidin conjugated with Alexa fluor for 1 h at room temperature and then washed using a washing solution for 3 h with a change of washing solution after every 30 min. The sensory epi-thelia are then mounted on the glass slide and 0.17 mm cover glass using aqueous mounting media

### Imaging

The sensory epithelia were imaged using Olympus Fluoview 3000 inverted microscope controlled by Olympus FV31S-SW software at Central Imaging and Flow Cytometry Facility (CIFF) at NCBS. The images were obtained using 60× oil immer-sion objective of numerical aperture (NA) 1.42. The start and end point of the confocal volume were decided with the

presence of stereocilia and junctions; images were obtained with a step size 0.5 µm and pixel size as per Nyquist sampling criteria. The laser power was reduced using a neutral density filter (ND filter) to less than 10% and tuned from 0.1% to 10% for each experiment, keeping the voltage values between 400 and 500 V, gain at 1 and offset between 2%–3%, images were obtained by line sequential scanning. The light path was chosen such that the primary dichroic mirror and the secondary dichroic mirror were the same for each scanning, and the emitted light was collected by a high-sensitivity spectral detector. For constructing the whole sensory epithelia images, the images were obtained using 10× air objective of NA 0.4 with step size of 1 µm, line sequential scanning, HV values ranging between 400 and 500 V.

The Olympus FV31S-SW software converts signals to 16-bit depth. The rotation feature of the software is used to align tissue wherever necessary.

### Ex vivo organ culture

To culture cochlea, we used a three-dimensional collagen droplet culture previously explained [61]). Briefly, we made a collagen matrix solution by mixing 400 µl of rat tail collagen I, 50 µl of 10× DMEM, 30 µl of 7.5% $NaHCO_3$, and 5 µl of HEPES using a pipette. Collagen mixture is poured as separate drops in 4-well dishes. The cochlea from the mouse embryo is dissected, and a single cochlea is kept in each drop. Once all the drops have received one cochlea each the 4-well plate is transferred to a 37 °C incubator maintaining 5% $CO_2$ for 5 min. The plate is then taken out and the 500 µl of culture media (1× DMEM supplemented with N2 and penicillin) is added using micropipettes and sterilized tips. The plate is then incubated at a 37 °C incubator maintaining 5% $CO_2$ for the required time. For small molecule perturbation, the solutions of the small molecule are added to the culture media of a well and treated as experiment, and the solvent of the compound are added to the culture media and treated as control. For blocking experiments, Bovine Serum Albumin (BSA) was used as a control. The concentration of small molecule inhibitors and blocking antibodies is presented in S4 Table of Supporting information.

### Image analysis

The confocal images obtained were opened in FIJI (NIH Image J) and processed to form single-channel and multichannel merged images. Images were provided with a scale bar using the metadata confocal file. To evaluate the morphological features of the cells in the epithelia we segmented the images using the Tissue-Analyzer plugin and used the segmented images to obtain values of **apical surface area, shape index, circularity, Feret angle, aspect ratio** using the auto-measure tool in Image J. We also obtained other morphological features of the epithelia

A. **Polarity:** The position of kinocilia is ascertained by the Beta-spectrin2 or Arl13b staining. The X, Y coordinates of the cilia position is determined using Image J and a vector is drawn from the centroid of HCs to the cilia position. This vector length (r) is normalized to the radius of cells and the angle ($\theta$) is with respect to the P-D axis ($\theta$). The cilia position is then plotted along with the calculation of the mean angle (denoted by the angle of the arrow) and circular standard deviation (denoted by the length of the arrow) using the custom-made script, which would be deposited on GitHub after the acceptance of this manuscript.

B. **4-Cell vertex calculations:** The segmented images in Tissue analyzer are processed to obtain the data for bonds using SQL command. The data for bond position and vertex coordinates are tabulated in MS Excel. Using an MS Excel macro, the vertex with more than 4 cells and the total number of vertices are computed.

C. **Width of the compartments:** For calculation of sensory compartment width a straight line is drawn in FIHI from the medial edge of IHC to the lateral edge of border cells. The medial sensory compartment is calculated from medial edge of IHC to lateral edge of pillar cells; the lateral sensory compartment is calculated from medial edge of OHC1 to lateral edge of border cells.

D. **Tortuosity:** Using the centroid of IHC a line was drawn connecting 10 IHC and its length (L) was calculated and the shortest distance between the IHC was calculated between the first and last IHC (l). The tortuosity was calculated as ratio of L/l.

E. **Axis of cell elongation:** We segmented cells of KO using Tissue analyzer software. We then imported the segmented image in FIJI and using the measurement option to calculate Feret angle and circularity. Feret angle calculates the angle between the long axis of cells and the left-right axis of the image. We then picked cells with circularity less than 0.8 and plotted their feret angle which represents the angle of elongation, using the rose stack plot.

## Visualization

Graphs are made using Prism-GraphPad. Rose stack plots representing the axis of elongation for KO cells are plotted using Oriana software from Kovach computing services. The images are assembled into figures using Inkscape.

## Statistics

We performed a two-tail unpaired $T$ test without assuming Gaussian distribution as the Mann–Whitney $T$ test to calculate the significance level using Prism-GraphPad software.

## Supporting information

**S1 Fig. Radial patterning of sensory and non-sensory domains is preserved during cochlear elongation. (A)** Schematic representing the organization of various sensory and non-sensory domains and their constituent cell types in the mouse organ of Corti. **(B)** Base of E15.5 and E18.5 OC stained for F-actin (green) and Myosin7a (magenta). **(C)** Base of E15.5 and E18.5 OC stained for F-actin (green) and p75NTR (magenta). **(D)** Base of E15.5 and E18.5 OC stained for F-actin (green) and BLBP (magenta). **(E)** OC from E15.5 and PO stained for Myosin 7a. Scale Bar: 50 μm in E and 5 μm in B–D. Image orientation: Top is lateral, Right is Apex.
(TIF)

**S2 Fig. Domain organization is maintained even at the apical turn in *looptail* mutants. (A)** OC from embryonic day (E)18.5 heterozygous (*Vangl2* $^{Lp/+}$) and homozygous (*Vangl2* $^{Lp/Lp}$) *looptail* mutant stained for f-actin (green). **(B)** Length of E18.5 cochlea from heterozygous (*Vangl2* $^{Lp/+}$) and homozygous (*Vangl2* $^{Lp/Lp}$) *looptail* mutant. $N=4/5$ (Het/Homo). **(C)** Apex of E18.5 OC from heterozygous (*Vangl2* $^{Lp/+}$) and homozygous (*Vangl2* $^{Lp/Lp}$) *looptail* mutant stained for F-actin (green) and Myosin 7a (magenta and gray). $N=4$. **(D)** Apex of E18.5 OC from heterozygous (*Vangl2* $^{Lp/+}$) and homozygous (*Vangl2* $^{Lp/Lp}$) *looptail* mutant stained for F-actin (green) and p75NTR (magenta and gray). $N=4$. **(E)** Apex of E18.5 OC from heterozygous (*Vangl2* $^{Lp/+}$) and homozygous (*Vangl2* $^{Lp/Lp}$) *looptail* mutant stained for F-actin (green) and BLBP (magenta and gray). $N=4$. Scale Bar: 50 μm in B and 5 μm in C–E. Unpaired $T$ test. Image orientation: Top is lateral, Right is Apex. Underlying data available in S1 Data
(TIF)

**S3 Fig. Differential expression of adhesion molecule super-impose on domain identity. (A)** E18.5 OC stained for F-actin (green) and Nectin2 (magenta and gray). **(B)** E18.5 OC stained for F-actin (green) and Zonula Occludens-1 (ZO1) (magenta and gray). **(C)** E18.5 OC stained for F-actin (green) and Nectin2 (magenta and gray). **(D)** E18.5 OC stained for F-actin (green) and Cdh1 (magenta and gray). **(E)** E18.5 OC stained for F-actin (green) and Cdh2 (magenta and gray). **(F)** E18.5 OC stained for F-actin (green) and Cdh4 (magenta and gray). **(G)** Schematic representing the combinatorial expression of adhesion molecule super-imposed on the cell types of OC. **(H)** E14.5 OC stained for F-actin (gray), Sox2 (magenta and gray), and Cdh1 (green and gray). Scale Bar: 10 μm. Image orientation: Top is lateral, Right is Apex.
(TIF)

**S4 Fig. Disruption of Fgfr1 signaling leads to sustained misexpression of cadherin and disruption in organization.** (A) E18.5 OC from control embryos (*Six1enh21Cre*$^{-/-}$*::Fgfr1*$^{fl/fl}$) stained for Cdh1 (gray and magenta), Cdh2 (gray and green), and F-actin (gray in merged). Asterisk indicates the absence of Cdh2 signals from lateral non-sensory domain. *N*=4 embryos. (B) E18.5 OC from Fgfr1 mutant embryos (*Six1enh21Cre*$^{+/-}$*::Fgfr1*$^{fl/fl}$) stained for Cdh1 (gray and magenta), Cdh2 (gray and green), and F-actin (gray in merged). Asterisk indicates the ectopic Cdh2 signals from lateral non-sensory domain. *N*=4 embryos. (C) Relative Fluorescence Intensity of Cdh2 in Claudius cells in control and Fgfr1 mutant cochlea at E18.5. (D) Number of IHC surrounded by KO cells on all side per cochlea in control and fgfr1 mutant cochlea. This shows medial non-sensory domain is intermixed with the medial sensory domain. (E) Number of OHC surrounded by Hensen's cells on all side per cochlea in control and fgfr1 mutant cochlea. This shows lateral non-sensory domain is intermixed with the lateral sensory domain. (F) E18.5 OC from control and Fgfr1 mutant embryos (*Six1enh21Cre*$^{+/-}$*::Fgfr1*$^{fl/fl}$) stained for F-actin (green) and HC marker Beta Spectrin II (magenta). Asterisk indicates the presence of HCs in the lateral and medial non-sensory domains. *N*=5 embryos. Unpaired *T* test. Scale Bar: 10 µm. Image orientation: Top is lateral, Right is Apex. Underlying data available in S1 Data.
(TIF)

**S5 Fig. Disruption of FGF signaling drives misexpression of Cdh2 in lateral non-sensory domain.** (A) Schematic representing the collagen droplet culture and treatment condition for an ex vivo explant culture of OC. (B) E16.5 Cochlea cultured in presence of DMSO for 12 h stained for F-actin (green), BLBP (magenta), and p75NTR (magenta). (C) E16.5 Cochlea cultured in presence of Fgfr1 inhibitor (Su5402, 25 µM) stained for F-actin (green), BLBP (magenta), and p75NTR (magenta). (D) E16.5 Cochlea cultured in presence of Fgfr1 inhibitor (Su5402, 25 µM) stained for F-actin (green), Cdh1(magenta). (E) E16.5 Cochlea cultured in presence of Fgfr1 inhibitor (Su5402, 25 µM) stained for F-actin (green), Cdh2 (magenta, gray) showing misexpression of Cdh2 in Claudius cells similar to the genetic perturbation in S4B Fig. (F) Relative Fluorescence Intensity of Cdh2 in Claudius cells in control (DMSO treated) and Fgfr1 inhibited (Su5402 treated) cochlea at E18.5. Unpaired *T* test. Scale Bar: 10 µm. Image orientation: Top is lateral, Right is Apex. Underlying data available in S1 Data.
(TIF)

**S6 Fig. FGF inhibited and cadherin blocked OC showed disorganization of domains.** (A) E16.5 Cochlea cultured in presence of DMSO and Su5402 for 12 h stained for F-actin (green), BLBP (magenta). (B) Fraction of HC-HC contacts in DMSO-treated and Su5402-treated OC. (C) Straightness of IHC represented by low tortuosity in DMSO-treated cochlea and the disruption of this straightness represented by high tortuosity in Su5402-treated cochlea. (D) Cochlea cultured for 1 h in presence of cadherin blocking antibodies 7D6, which block interactions among Cdh1, stained using F-actin (green) and secondary antibodies (magenta, gray). (E) Cochlea cultured for 1 h in presence of cadherin blocking antibodies 6B3, which block interactions among Cdh2, stained using F-actin (green) and secondary antibodies (magenta, gray). (F) 12-h explant of E15.5 OC in presence of Bovine Serum Albumin (BSA, 0.1%), Cdh1 blocking antibodies (7D6, 10 µg/ml), Cdh2 blocking antibodies (6B3, 10 µg/ml) stained for F-actin (green) and Arl13b (magenta). *N*=4 cochlea. (G) Fraction of HC-HC contacts in BSA-treated, Cdh1-blocked, and Cdh2-blocked OC. (H) Tortuosity of IHC in BSA-treated, Cdh1 blocked and Cdh2-blocked OC. Unpaired *T* test. Scale Bar: 10 µm and Image orientation: Top is lateral, Right is Apex. Underlying data available in S1 Data.
(TIF)

**S7 Fig. Adhesion code is maintained in *looptail* mutants with defective convergent extension.** (A) E18.5 OC from heterozygous (*Vangl2*$^{Lp/+}$) and homozygous (*Vangl2*$^{Lp/Lp}$) *looptail* mutant stained for F-actin (green) and Cdh1 (magenta and gray). *N*=3 cochlea. (B) E18.5 OC from heterozygous (*Vangl2*$^{Lp/+}$) and homozygous (*Vangl2*$^{Lp/Lp}$) *looptail* mutant stained for F-actin (green) and Cdh2 (magenta and gray). *N*=3 cochlea. (C) E18.5 OC from heterozygous (*Vangl2*$^{Lp/+}$) and homozygous (*Vangl2*$^{Lp/Lp}$) *looptail* mutant stained for F-actin (green) and Cdh4 (magenta and gray). *N*=3 cochlea. (D) E18.5 OC from

heterozygous (*Vangl2 $^{Lp/+}$*) and homozygous (*Vangl2 $^{Lp/Lp}$*) *looptail* mutant stained for F-actin (green) and Nectin1 (magenta and gray). *N*=3 cochlea. **(E)** Relative fluorescence intensity of Cdh1, Cdh2, and Cdh4 along the medio-lateral axis of OC at E18.5 from heterozygous (*Vangl2 $^{Lp/+}$*) and homozygous (*Vangl2 $^{Lp/Lp}$*) *looptail* mutant in pink and green, respectively. Scale Bar: 10 μm. Image orientation: Top is lateral, Right is Apex. Underlying data available in S1 Data
(TIF)

**S8 Fig. Adhesion code ensures discrete organization of each compartment. (A)** Schematic of OC representing the nine equidistant points along the base-apex axis of the OC. **(B)** Width of the lateral sensory domain along the nine positions along the base-apex axis at E15.5 and E18.5. *N*=4 cochlea each stage. **(C)** Width of the medial sensory domain along the nine positions along the base-apex axis at E15.5 and E18.5. *N*=4 cochlea each stage. **(D)** Shape index (*Q*=perimeter/sqrt of area) of Hensen's Cells at E16.5 and E18.5. *N*=150/169 for Hensen and 144/153 for Claudius (E16.5/E18.5). **(E)** Shape index (*Q*=perimeter/sqrt of area) of Claudius Cells at E16.5 and E18.5. *N*=144/153 (E16.5/E18.5). **(F)** E18.5 base of OC from control (*Six1enh21Cre$^{-/-}$::Fgfr1$^{fl/fl}$*) and Fgfr1 mutant embryos (*Six1enh21Cre$^{+/-}$::Fgfr1$^{fl/fl}$*) stained for F-actin (gray) showing KO domain. *N*=4. **(G)** Schematic representing the calculation of the axis of cell elongation in mKO and lKO cells of KO domain. **(H)** Apical surface area of mKO and lKO cells from control (*Six1enh21Cre$^{-/-}$::Fgfr1$^{fl/fl}$*) and Fgfr1 mutant embryos (*Six1enh21Cre$^{+/-}$::Fgfr1$^{fl/fl}$). N*=150/216 for control and 209/262 for mutant (mKO/lKO). **(I)** Circularity of mKO and lKO cells from control (*Six1enh21Cre$^{-/-}$::Fgfr1$^{fl/fl}$*) and Fgfr1 mutant embryos (*Six1enh21Cre$^{+/-}$::Fgfr1$^{fl/fl}$). N*=150/216 for control and 209/262 for mutant (mKO/lKO). **(J)** E18.5 base of OC from control (*Six1enh21Cre$^{-/-}$::Fgfr1$^{fl/fl}$*) and Fgfr1 mutant embryos (*Six1enh21Cre$^{+/-}$::Fgfr1$^{fl/fl}$*) stained for F-actin (gray) showing lateral non-sensory domain. *N*=4. **(K)** Apical surface area of Hensen's Cells from control (*Six1enh21Cre$^{-/-}$::Fgfr1$^{fl/fl}$*) and Fgfr1 mutant embryos (*Six1enh21Cre$^{+/-}$::Fgfr1$^{fl/fl}$). N*=65/98. (control/mutant). **(L)** Shape index of Hensen's Cells from control (*Six1enh21Cre$^{-/-}$::Fgfr1$^{fl/fl}$*) and Fgfr1 mutant embryos (*Six1enh21Cre$^{+/-}$::Fgfr1$^{fl/fl}$*). *N*=65/98. (control/mutant). Scale Bar: 10 μm. Unpaired *T* test. Image orientation: Top is lateral, Right is Apex. *Six1enh21Cre$^{-/-}$* means cre negative and *Six1enh21Cre$^{+/-}$* cre positive. Underlying data available in S1 Data.
(TIF)

**S9 Fig. Cellular intercalation drives compartment-specific reorganization. (A)** Schematic representing the timeline of EdU injection and staining. **(B)** OC of embryos from E13.5, E15.5, E16.5, and E18.5 pregnant females injected with EdU stained for Sox2 (green) to mark sensory epithelia and click-chemistry based EdU (Magenta). *N*=3. **(C)** OC from E15.5 and E18.5 base position (30P) stained for F-actin (gray), overlayed with red circles representing vertex with 4 or more cells. **(D)** Percentage of vertex with 4 or more cells in the sensory domain at E15.5 and E18.5. *N*=3 embryos each with 205/615 for E15.5 and 122/625 for E18.5 (4 or more cell vertices/Total vertices). **(E)** Percentage of vertex with 4 or more cells in the medial and lateral KO domain at E15.5 and E18.5. *N*=3 cochlea each, 498/1070 mKO at E15.5 and 259/630 at E18.5; 318/1532 lKO at E15.5 and 111/835 at E18.5 (4 or more cell vertices/Total vertices). Scale Bar: 20 μm for E13.5 and 10 μm for rest. Unpaired *T* test, ns=*P*>0.05. Image orientation: Top is lateral, Right is Apex. Underlying data available in S1 Data.
(TIF)

**S10 Fig. NMII-activity drives organization within each compartment. (A)** E18.5 OC stained for F-actin (green) and mono-phosphorylated form of RLC (pRLC, magenta, gray). **(B)** KO and Sensory domain from E18.5 OC stained for ZO1(green) and di-phosphorylated form of RLC (ppRLC, magenta, gray). **(C)** E16.5 cochlea cultured ex vivo for 8h in a 3D-collagen droplet culture with DMEM supplemented with DMSO, stained for F-actin (green) and Arl13b (magenta). *N*=6 cochlea. **(D)** E16.5 cochlea cultured ex vivo for 8h in a 3D-collagen droplet culture with DMEM supplemented with or 25 μM ML7, stained for F-actin (green) and Arl13b (magenta). *N*=6 cochlea. **(E)** Polar coordinates representing position of kinocilia of IHC from OC cultured in presence (blue) or absence (orange) of MLCK-inhibitor ML7. *N*=66/68 (DMSO/ML7). **(F)** Polar coordinates representing position of kinocilia of OHC from OC cultured in presence (blue) or absence (orange) of MLCK-inhibitor ML7. *N*=68/61 (DMSO/ML7). **(G)** Circularity of IHC and OHC from OC cultured in presence (blue) or absence (orange) of MLCK-inhibitor ML7. *N*=58/61 for IHC and 69/66 for OHC (DMSO/ML7). **(H)** Apical surface area of

IHC and OHC from OC cultured in presence (blue) or absence (orange) of MLCK-inhibitor ML7. $N=58/61$ for IHC and 69/66 for OHC (DMSO/ML7). **(I)** Apical surface area of mKO and lKO cells from OC cultured in presence or absence of MLCK-inhibitor ML7. $N=193/207$ for DMSO and 203/214 for ML7 (DMSO/ML7). **(J)** Rose stack plot representing the axis of cell elongation for mKO and lKO cells in OC cultured in DMSO. $N=193/207$ mKO/lKO. **(K)** Rose stack plot representing the axis of cell elongation for mKO and lKO cells in OC cultured in ML7. $N=203/214$ mKO/lKO. Image orientation: Top is lateral, Right is Apex. Scale Bar: 10 μm. Unpaired $T$ test, ns = non-signficant, $P>0.05$. Underlying data available in S1 Data.
(TIF)

**S11 Fig. Vangl2-driven NMII-activity drives domain-scale and cellular-scale organization. (A)** Sensory domain from E18.5 OC stained for F-actin (green) and Vangl2 (magenta). **(B)** Sensory domain from E18.5 OC stained for ZO1(green) and di-phosphorylated form of RLC (ppRLC, magenta, gray). **(C)** Base region (30P) of E18.5 OC from heterozygous (*Vangl2 $^{Lp/+}$*) and homozygous (*Vangl2 $^{Lp/Lp}$*) *looptail* mutant stained for F-actin (green) and Arl13b (magenta). $N=6$ cochlea. **(D)** Polar coordinates of kinocilia of IHC and OHC at Base region (30P) of E18.5 OC from heterozygous (*Vangl2 $^{Lp/+}$*) in orange and homozygous (*Vangl2 $^{Lp/Lp}$*) *looptail* mutant in blue. $N=66/66$ for IHC and $N=110/110$ for OHC (Het/Homo). **(E)** E18.5 OC from heterozygous *looptail* mutant stained for F-actin (green) and pRLC (magenta, gray). **(F)** E18.5 OC from homozygous *looptail* mutant stained for F-actin (green) and pRLC (magenta, gray). **(G)** E18.5 OC from heterozygous *looptail* mutant stained for ZO1(green) and ppRLC (magenta, gray). **(H)** E18.5 OC from homozygous *looptail* mutant stained for ZO1(green) and ppRLC (magenta, gray). **(I)** Width of medial and lateral sensory domain along the OC from heterozygous (*Vangl2 $^{Lp/+}$*) and homozygous (*Vangl2 $^{Lp/Lp}$*) *looptail* mutant at E18.5. $N=4$ embryos. **(J)** Width of Kölliker's organ along the OC from heterozygous (*Vangl2 $^{Lp/+}$*) and homozygous (*Vangl2 $^{Lp/Lp}$*) *looptail* mutant at E18.5. $N=4$ embryos. Image orientation: Top is lateral, Right is Apex. Scale Bar: 10 μm. Unpaired $T$ test, ns = non-signficant, $P>0.05$, $*=P<0.05$. Underlying data available in S1 Data
(TIF)

**S12 Fig. Compartment-specific deletion of Vangl2 has non-linear effects on other compartments. (A)** Base region (30P) of E16.5 OC from Cre negative control (*Ngn1$^{457}$-Cre$^{-/-}$:: Vangl2$^{lp/fl}$*) and Ngn1$^{457}$cre positive mutant (*Ngn1$^{457}$-Cre$^{+/-}$:: Vangl2$^{lp/fl}$*) stained for F-actin (green) and Arl13b (magenta). **(B)** Base region (30P) of E18.5 OC from Cre negative control (*Ngn1$^{457}$-Cre$^{-/-}$:: Vangl2$^{fl/fl}$*) and Ngn1$^{457}$cre positive mutant (*Ngn1$^{457}$-Cre$^{+/-}$:: Vangl2$^{fl/fl}$*) stained for F-actin (green) and Arl13b (magenta). Note: here both copies of Vangl2 is flox allele. **(C)** E18.5 OC from Lgr5CreERT2::Ai14, induced with Tamoxifen stained with F-actin (green) showing cre-mediated expression of Tdtomato (magenta). **(D)** Percentage of tdtomato positive cells from the Ngn1$^{457}$-Cre::Ai14 cochlea in KO, sensory and lateral non-sensory domain. $N=3$. Scale Bar: 10 μm. Image orientation: Top is lateral, Right is Apex. Underlying data available in S1 Data.
(TIF)

**S13 Fig. Interaction between Vangl2 and Vinculin governs epithelial mechanics. (A)** Base region of E15.5 OC stained for F-actin (green) and Vinculin (magenta and gray), showing its localization on cell junctions of sensory and non-sensory compartments. **(B)** Base region of E18.5 OC stained for F-actin (green) and Vinculin (magenta, gray and heat map) showing higher localization of Vinculin on Hensen's cells and the lKO cells. **(C)** Base region of E18.5 OC from heterozygous (*Vangl2 $^{Lp/+}$*) and homozygous (*Vangl2 $^{Lp/Lp}$*) *looptail* mutant stained for F-actin (green) and Vinculin (magenta, gray and heat map). **(D)** Relative Fluorescence Intensity of Vinculin in cells of sensory and non-sensory compartments at E18.5 OC from heterozygous (*Vangl2 $^{Lp/+}$*) and homozygous (*Vangl2 $^{Lp/Lp}$*) *looptail* mutant. $N=30$ junctions from each cell type and genotype. **(E)** Width of Kölliker's organ along the OC from the control (*Emx2-Cre$^{-/-}$:: Vinculin $^{fl/fl}$*) and mutant for vinculin (*Emx2-Cre$^{+/-}$:: Vinculin $^{fl/fl}$*) at E18.5. $N=3$ embryos. Scale Bar: 10 μm. Image orientation: Top is lateral, Right is Apex. Unpaired $T$ test, ns = non-signficant, $P>0.05$, $*=P<0.05$. $****=P<0.0001$. Underlying data available in S1 Data.
(TIF)

**S1 Table. Mouse strains used for experiments in this paper.**
(PDF)

**S2 Table. Oligonucleotide sequences used for PCR-based genotyping mouse strains.**
(PDF)

**S3 Table. Primary and secondary antibodies used with the fixation condition.**
(PDF)

**S4 Table. Concentration of small molecule inhibitors used for ex vivo organ culture.**
(PDF)

**S1 Data. Underlying data for quantifications presented.**
(XLSX)

## Acknowledgments

We thank Central Imaging and Flowcytometry Facility (CIFF), Animal Care and Resource Centre (ACRC), and laboratory kitchen at NCBS. We thank Srikala Raghvan for providing the Vinculin mouse. We thank Hiroshi Hamada and Sandeep Krishna for comments on the manuscript. A. P. thanks ICTP—ICTS Winter School on Quantitative Systems Biology for introducing quantitative morphogenesis. The Myo7a antibody developed by Dana J. Orten, E-cadherin antibody developed by Warren Gallin, and the N-cadherin antibody developed by Karen A Knudsen was obtained from the Developmental Studies Hybridoma Bank, created by the NICHD of the NIH and maintained at The University of Iowa, Department of Biology, Iowa City, IA 52242. We are grateful to Sweety Meel and the members of Ear Lab for their feedback.

## Author contributions

**Conceptualization:** Anubhav Prakash, Raj K. Ladher.

**Data curation:** Anubhav Prakash, Raj K. Ladher.

**Formal analysis:** Anubhav Prakash, Anton S. Iyer, Raj K. Ladher.

**Funding acquisition:** Raj K. Ladher.

**Investigation:** Anubhav Prakash, Sukanya Raman, Raman Kaushik.

**Methodology:** Anubhav Prakash, Sukanya Raman, Raman Kaushik, Pallavi Manchanda, Anton S. Iyer, Raj K. Ladher.

**Project administration:** Raj K. Ladher.

**Supervision:** Raj K. Ladher.

**Validation:** Anubhav Prakash.

**Writing – original draft:** Anubhav Prakash, Raj K. Ladher.

**Writing – review & editing:** Sukanya Raman, Raman Kaushik, Pallavi Manchanda, Raj K. Ladher.

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
