## [Editor Report · Decision Letter 0]

14 Feb 2025

Dear Dr Ladher, 

Thank you for submitting your manuscript entitled "Compartment coupling integrates patterning and morphogenetic information during development" for consideration as a Research Article by PLOS Biology.

Your manuscript has now been evaluated by the PLOS Biology editorial staff as well as by an academic editor with relevant expertise and I am writing to let you know that we would like to send your submission out for external peer review.

Once your full submission is complete, your paper will undergo a series of checks in preparation for peer review. After your manuscript has passed the checks it will be sent out for review. To provide the metadata for your submission, please Login to Editorial Manager (https://www.editorialmanager.com/pbiology) within two working days, i.e. by Feb 17 2025 11:59PM.

Kind regards,

Ines

--

Ines Alvarez-Garcia, PhD

Senior Editor

PLOS Biology

---

## [Decision Letter · Decision Letter 1]

12 Apr 2025

Dear Dr Ladher,

Thank you for your patience while your manuscript entitled "Compartment coupling integrates patterning and morphogenetic information during development" was peer-reviewed at PLOS Biology. The manuscript has now been evaluated by the PLOS Biology editors, an Academic Editor with relevant expertise, and by three independent reviewers. 

The reviews are attached below. As you will see, the reviewers find the conclusions novel and interesting, but they also raise several issues that would need to be addressed before we can consider. Both Reviewer 1 and 3 ask for clarifications in the figures and several points on the text. Reviewer 2 thinks you should improve the writing and figures, as some of the results are difficult to assess, and also mentions that a detailed description on how the axis of elongation for KO cells was measured should be provided along with the clarification of several issues.

In light of the reviews, we would like to invite you to revise the work to thoroughly address the reviewers' reports. Given the extent of revision needed, we cannot make a decision about publication until we have seen the revised manuscript and your response to the reviewers' comments. Your revised manuscript is likely to be sent for further evaluation by all or a subset of the reviewers.

**IMPORTANT - SUBMITTING YOUR REVISION**

3. Resubmission Checklist

a) *PLOS Data Policy*

b) *Published Peer Review*

Sincerely,

Ines

--

Ines Alvarez-Garcia, PhD

Senior Editor

PLOS Biology

Reviewers' comments

Rev. 1:

How part of the epithelium in the inner ear especially the cochlea adopts a sensory fate is unknown. It is clear that there is patterning of the various cell types and this patterning medial to lateral is maintained during convergent extension in the longitudinal direction. This manuscript proposes that specific cell adhesion proteins allow cells in a particular domain to remain together. The authors describe a combinatorial code of adhesion molecules defining specific compartments in the developing cochlea. They further elucidated using Fgfr1 conditional Knockouts that Fgfr1 was essential for maintaining the code for the domains. They further show that Fgfr1 activity is necessary even after cell fate choices are made. Disruption of Cdh signaling with specific blocking antibodies show that these adhesion molecules were necessary for domain segregation and that blocking of specific cadherins led to disruption of particular domains. In an interesting experiment deleting Vangl2 in separate compartments showed that each compartment can influence one another. This is a novel finding, a phenomenon they refer to as "compartment coupling". Testing a possible mechanism the authors propose a mechanical force influence and indeed when Vinculin, a protein required for transmission of forces across junctions, is deleted coupling between compartments is disrupted.

There are many studies describing radial patterning of the cochlea and its dependence on various secreted factors. Not so much is known about the maintenance of patterning during the convergent extension required for the cochlea to reach its full length. This study introduces some new players consisting of planar cell polarity factors as well as cell adhesion molecules. It provides an interesting and new perspective for developmental biologists studying the cochlea.

Specific suggestions to enhance clarity:

In Figure 1 panel B which is labeled E15.5 looks much older since at E15.5 the cochlea is not well patterned.

Make graph on Fig S12 larger i.e. the size of the image panels of the same figure.

Figure 4: truncate panel A and enlarge panel B

Rev. 2:

This manuscript described pattern formation of developing organ of Corti focusing on Kollikers organ, medial non-sensory domain. They identified that organization of these domains established before the sensory epithelium specification, FGF and PCP signals are required for patterning, domain organization is interconnected, and compartment coupling was mediated via vinculin-dependent junctional mechanics.

The mechanism by which pattern development and coupling through junctional mechanics is interesting and this will be the topic of interest in cell biology. However, there are concerns on this manuscript need to be revised before publication.

1. The manuscripts contain many typos and mis-labeling figures which are difficult to understand and interpret. For example, in Figure S3G, lKO region (light brown) was indicated as Cdh 1+2+4 positive but it looked like Cdh-2+4 positive based on other subfigures. Lateral-sensory region (pink) also seem to be labeled by Cdh1+Nectrin1 instead of Cdh2. Lateral non-sensory region (grey) seems Cdh1 only not Cdh1+Nectrin1.

2. In general, the quality of the pictures is not optimal to assess what the authors claim. Major concern is that the phalloidin staining in the KO region, especially mKO region is hard assess (Fig3G, Fig4C, Fig5F, FigS8F). This may be due to flattening the KO region since KO region is composed of multilayer increasing from lKO to mKO.

3. There is not detailed description how to measure axis of elongation for KO cells. This makes it difficult to understand Rose stack plots.

4. In Figure2, it is difficult to assess the expression pattern of Cdh1, 2 and Phalloidin. Inner hair cells are supposed to specify at E14.5 by expressing Atoh1 and Sox2, Jag1, and are supposed to be expressed in the organ of Corti. It would be better to co-label any of those with phalloidin and cadherins to visualize sensory patch. Cdh2 staining in panel G was not convincing as plotted on the RFI on panel H. The authors said "the patterned expression of Cdh1 and Cdh2 OC was absent" in Fgfr1-cko. This statement needs to be more specific. Are Cdh1 and Cdh2 expressed in the same cells or in a mosaic pattern? Since Chd1 and Chd2 are not co-expressed in the control OC, this would be important to know the exact expression pattern.

5. In Figure 3G, scale of control and Vangl2Lp/Lp panel is different. This need to be fixed to for readers to compare easily the area of cells.

6. Data in Figure 6 is surprising. Only 10% of cells within the KO domain were targeted in Ngn1-457Cre. Yet, degree of the phenotype in Vangl2-cKO are comparable to Vangl2Lp/Lp and Vangl2-Lgr5-cKO. This is dramatic effect of Vangl2 in patterning of OC. Authors need to discuss this phenomenon.

7. The phenotype of Vinclulin-cKO is different from Vangl2-KO, increased IKO surface area instead of decrease and different mKO long axis direction. Authors need to explain this difference.

Rev. 3:

The findings presented in this study show that cell patterning in the developing mouse cochlea is associated with a combinatorial expression of adhesion molecules, which segregates the OC into spatially defined compartments and allow planar cell polarity (PCP) cues to regulate cell organisation within each compartment. The work implicates compartment coupling as a major determining factor in cochlear morphogenesis. Although we understand a significant amount about how the cells within the cochlear epithelium are specified, the mechanisms underlying its growth and morphogenesis are less clear. This is therefore a novel study that addresses an under explored and important aspect of inner ear development.

The authors use different mouse mutants to understand how compartment-coupling and compartment integrity is maintained during convergent extension in the developing cochlea. They describe compartment-coupling as mechanism where a non-linear influence on the organisation in one compartment can disrupt cellular organisation and patterning in another. Overall, this compartment-coupling is regulated by PCP signalling via Vangl2.

Given that Vangl2 regulates junctional mechanics between different cells, the authors explored a potential mechanical component of compartment coupling.

Questions to authors:

The authors describe a change in cell shape index and circularity in the Ffgr1 mutants. Do the authors think this could be due to changes in the cytoskeleton or cell-specific tension? Were either of these parameters investigated or compared between WT and mutant cochleae? Would it be possible to infer any hypothesis about how epithelial mechanics might between cochlear compartments? Are there differences in the expression levels of vinculin in different compartments?

The differences in the direction of the long axis of lKO and mKO cells are reduced in Emx2-Cre+/- ::Vincfl/f 'These data suggest the coupling between compartments is mediated, at least in part, through mechanics.'

From this statement, would the hypothesis assume that cells are smaller because the are under less tension, or because they are being compressed from changes in cell properties/behaviour in another compartment where vinculin activity has been lost?

Were there any differences in gross cochlear length in the absence of vinculin?

Were there any differences in the OC width? If mechanical coupling has been altered, one could hypothesise potential changes in compartment size. Was this looked at?

Is it possible to provide data on normal expression of vinculin in the developing cochlea?

The authors describe a loss of circularity in a decrease in apical surface area in OHCs following treatment with ML7(Fig. S10C-H), This finding is attributed to a decrease in spatial organisation within the sensory domain. How can the authors be sure this is not simply a reduction in overall HC growth or developmental progression?

Given the reported role for vinculin in compartment coupling do the authors think there is any interaction between Vangl2 and vinculin in terms of regulating cell behaviour or cell tension across different compartments. What would be the predicted outcome of reduced vinculin in a Vangl2 LoF model? How would this impact compartment coupling? Could the authors provide any comment on this?

In Figure 5A the hair cells in the Lgr5CreERT2 +/-:: Vangl2Lp/fl::Tamox explants appear larger. Is this a consistent observation?

'Moreover, mutant cochlea showed no difference between the direction of the long axis of lKO cells (mediolateral to the tissue axis) and that of the mKO cells (parallel to the tissue axis) (Fig. 5F, H)'

In Figure 5F the directionality/orientation of the KO cells appears different along the medial-to-lateral axis in the Lgr5CreERT2 +/-:: Vangl2Lp/fl::Tamox compared to Lgr5CreERT2 +/-:: Vangl2Lp/fl::Corn oil. The cells in Lgr5CreERT2 +/-:: Vangl2Lp/fl::Corn oil appear polarised along the basal-to-apical axis whereas those in the Lgr5CreERT2 +/-:: Vangl2Lp/fl::Tamox appear polarised along medial-to-lateral axis. Can the authors provide further clarification on this?

'This data suggests a reciprocal influence of the sensory domain on cellular organisation within the KO compartment (Fig. 5I)………'

Could the authors provide further explanation on how they arrived at this conclusion. It is not immediately clear from the current text.

Could the authors comment on whether any differences in total cochlear length were observed between Lgr5CreERT2 +/-:: Vangl2Lp/fl::Corn oil and Lgr5CreERT2 +/-:: Vangl2Lp/fl::Tamox?

Can the authors provide any insight as to how compartment coupling might regulate gross cochlear morphogenesis or epithelial bending? How might coordinated signalling between Fgf and ERK described by Ishii et al in 2020 be impacted in the different compartment-specific Vangl2 LoF cochleae?

Similarly, how might cellular organisation within compartments be impacted following loss of Shh signalling or in other CE mutants not associated with components the core PCP pathway?

Minor points:

Check that all referencing is presented in the same format. Some references are 'numbered' and some refer to the author names. Huh et al 2011 and Ono et al 2014 are missing from the bibliography.

I recommend a thorough check of the final document for any remaining minor spelling/typographical errors.

---

## [Decision Letter · Decision Letter 2]

4 Jul 2025

Dear Dr Ladher,

Thank you for your patience while we considered your revised manuscript entitled "Compartment coupling integrates patterning and morphogenetic information during development" for publication as a Research Article at PLOS Biology. This revised version of your manuscript has been evaluated by the PLOS Biology editors, the Academic Editor and two of the original reviewers.

Based on the reviews, we are likely to accept this manuscript for publication, provided you satisfactorily address the policy-related requests stated below my signature.

In addition, we would like you to consider a suggestion to improve the title:

“Spatial Compartmentalization Drives Morphogenetic Patterning in the Organ of Corti”

We expect to receive your revised manuscript within two weeks. 

*Published Peer Review History*

*Press*

Sincerely,

Ines

--

Ines Alvarez-Garcia, PhD

Senior Editor

PLOS Biology

ETHICS STATEMENT:

Thank you for providing the ethics statement, but please also include an approval number.

Fig. 1C; Fig. 2E, H; Fig. 3B, C, D, F, H-J; Fig. 4B, D, E, G-J; Fig. 5B, D, E, G, H, I; Fig. 6C-F, H, I; Fig. S2B; Fig. S4C, D, E; Fig. S5F; Fig. S6B, C, G, H; Fig. S7E; Fig. S8B-E, H, I, K, L; Fig. S9D, E; Fig. S10E-K; Fig. S11I, J; Fig. S12D and Fig. S13D, E

CODE POLICY

Reviewers' comments

Rev. 2:

The authors responded the reviewers concerns well and the manuscript has been improved as a high quality one.

Rev. 3: Zoe Mann - note that this reviewer has signed the review

The authors have addressed all concerns and responded to all queries.

---

## [Editor Report · Decision Letter 3]

4 Aug 2025

Dear Dr Ladher,

Thank you for the submission of your revised Research Article entitled "Coupling between spatial compartments integrates morphogenetic patterning in the organ of Corti" for publication in PLOS Biology. On behalf of my colleagues and the Academic Editor, Alan Cheng, I am delighted to let you know that we can in principle accept your manuscript for publication, provided you address any remaining formatting and reporting issues. These will be detailed in an email you should receive within 2-3 business days from our colleagues in the journal operations team; no action is required from you until then. Please note that we will not be able to formally accept your manuscript and schedule it for publication until you have completed any requested changes.

PRESS

Sincerely, 

Ines

--

Ines Alvarez-Garcia, PhD

Senior Editor

PLOS Biology
